# Beyond Point-wise Neural Collapse: A Topology-Aware Hierarchical Classifier for Class-Incremental Learning

**Huiyu Yi** [1 2]  **Zhiming Xu** [1 2]  **Dunwei Tu** [1 2]  **Zhicheng Wang** [1 3]  **Baile Xu** [1 2]  **Furao Shen** [1 2]

## Abstract

The Nearest Class Mean (NCM) classifier is widely favored in Class-Incremental Learning (CIL) for its superior resistance to catastrophic forgetting compared to Fully Connected layers. While Neural Collapse (NC) theory supports NCM's optimality by assuming features collapse into single points, non-linear feature drift and insufficient training in CIL often prevent this ideal state. Consequently, classes manifest as complex manifolds rather than collapsed points, rendering the single-point NCM suboptimal. To address this, we propose Hierarchical-Cluster SOINN (HC-SOINN), a novel classifier that captures the topological structure of these manifolds via a "local-to-global" representation. Furthermore, we introduce Structure-Topology Alignment via Residuals (STAR) method, which employs a fine-grained pointwise trajectory tracking mechanism to actively deform the learned topology, allowing it to adapt precisely to complex non-linear feature drift. Theoretical analysis and Procrustes distance experiments validate our framework's resilience to manifold deformations. We integrated HC-SOINN into seven state-of-the-art methods by replacing their original classifiers, achieving consistent improvements that highlight the effectiveness and robustness of our approach. Code is available at https://github.com/yhyet/HC_SOINN.

---

[1]National Key Laboratory for Novel Software Technology, Nanjing University, Nanjing, China [2]School of Artificial Intelligence, Nanjing University, Nanjing, China [3]School of Computer Science, Nanjing University, Nanjing, China. Correspondence to: Baile Xu <xubaile@nju.edu.cn>.

*Proceedings of the 43rd International Conference on Machine Learning*, Seoul, South Korea. PMLR 306, 2026. Copyright 2026 by the author(s).

## 1. Introduction

Class-Incremental Learning (CIL) aims to enable models to learn new categories sequentially without catastrophically forgetting previously acquired knowledge (Belouadah et al., 2021; Zhou et al., 2024a; McCloskey & Cohen, 1989). While traditional Fully Connected (FC) classifiers are highly susceptible to forgetting due to severe weight bias towards new tasks (Nguyen et al., 2019), the Nearest Class Mean (NCM) classifier (Rebuffi et al., 2017) has emerged as a cornerstone of modern CIL frameworks owing to its superior robustness. This empirical preference has gained significant theoretical traction through the lens of Neural Collapse (NC) (Papyan et al., 2020; Yang et al., 2023). NC theory proves that during the terminal phase of training, intra-class features collapse into their respective class means, rendering the NCM classifier equivalent to an optimal linear classifier. Consequently, NCM-based approaches have become the de facto standard for maintaining feature-classifier alignment in incremental scenarios (Zhou et al., 2025; Yi, 2024; Zhu et al., 2024; Tu et al., 2025b).

However, the assumption that models reach this ideal terminal phase is frequently violated in practical CIL settings due to three critical factors. First, models in the initial task are often not trained to the point of collapse to avoid severe overfitting. Second, during incremental steps, training on new classes is typically constrained (e.g., via regularization or distillation) to preserve old class performance, leading to insufficient convergence. Finally, the pervasive "feature drift" problem causes the representations of old classes to shift as the backbone updates (Yu et al., 2020), further distancing the model from the collapsed state. Under these constraints, the NC1 property (zero intra-class variance) is compromised (Papyan et al., 2020). Class features no longer collapse into single points but instead reside on complex, high-dimensional manifolds. While NCM is robust to simple linear translations, it fails to represent these manifolds when they undergo non-linear drift, potentially forming "dumbbell" or "crescent" structures that deviate from the single-prototype assumption (Allen et al., 2019).

In this paper, we challenge the reliance on single-prototype classifiers and propose a topology-aware representation for CIL. We introduce **HC-SOINN** (Hierarchical-Cluster

SOINN), a novel classifier designed to capture the intrinsic structure of class manifolds. By integrating hierarchical clustering with an improved Self-Organizing Incremental Neural Network (SOINN)(Furao & Hasegawa, 2006), HC-SOINN represents each category through a combination of "local centers" and a "global center." During inference, these topological points are fused to provide a decision boundary that more faithfully respects the manifold's geometry.

Furthermore, our Procrustes analysis reveals that feature drift in CIL is far from a simple rigid transformation (Goldberg & Ritov, 2009), with normalized distances frequently reaching $0.3 - 0.4$ in later tasks. This significant deviation indicates that the feature drift is predominantly non-linear, involving complex deformations that cannot be approximated by simple rigid transformations. Although the proposed HC-SOINN, with its topological representation, is inherently more resilient to such deformations than the single-prototype NCM, its capacity to withstand severe non-linear distortion remains finite. To bridge this gap, we propose **STAR** (Structure-Topology Alignment via Residuals). Leveraging the multi-node granularity of HC-SOINN, STAR moves beyond global alignment to perform flexible, pointwise trajectory tracking. This mechanism enables the learned manifold structure to actively deform and precisely adapt to the complex evolution of the feature space.

Our contributions are summarized as follows:

- We revisit NCM's optimality under Neural Collapse and use Procrustes analysis to empirically reveal that feature drift causes significant non-linear structural deformations, exposing the failure of single-prototype methods.

- We propose the HC-SOINN classifier, which utilizes a hierarchical topological structure to represent class manifolds more accurately than single-prototype methods.

- We introduce STAR to shift the paradigm from drift resistance to drift adaptation. Leveraging pointwise trajectory tracking, STAR enables the topology to actively deform to match complex non-linear feature evolution.

- Extensive experiments on three mainstream CIL benchmarks and integration with seven state-of-the-art methods demonstrate the effectiveness and robustness of our approach.

## 2. Related Works

**Continual Learning with Pre-trained Models** leverages the robust representations of PTMs to mitigate forgetting (Zhou et al., 2024a). Parameter-Efficient Fine-Tuning (PEFT) approaches, such as DualPrompt (Wang et al., 2022b), CODA-Prompt (Smith et al., 2023), MQMK (Tu et al., 2025a), SEMA (Wang et al., 2025), and CL-LoRA (He et al., 2025), adapt backbones through task-specific prompts, attention-based weighted summation, or modularized adapters and LoRA modules. Alternatively, representation-based methods focus on exploiting frozen features; for instance, SimpleCIL (Zhou et al., 2025) uses class prototypes as baselines, while APER and EASE (Zhou et al., 2024b) further enhance performance via embedding aggregation or expandable subspace ensembles. Beyond these, KAC introduces a non-linear classifier based on Kolmogorov-Arnold networks to refine decision boundaries and mitigate drift (Hu et al., 2025).

**Neural Collapse Theory**

Neural Collapse theory describes a phenomenon observed during the terminal phase of training, mathematically formalized through four key properties (Papyan et al., 2020): Variability Collapse (NC1), where intra-class feature variance approaches zero as samples converge to their class means $\mu_k$; Convergence to Simplex ETF (NC2), where these class means form a geometrically optimal Simplex Equiangular Tight Frame; Feature-Classifier Alignment (NC3), implying classifier weights align perfectly with the normalized class means; and Simplification to NCC (NC4), where the decision rule becomes equivalent to the NCM classifier. In the context of Few-Shot Class Incremental Learning (FSCIL) (Tao et al., 2020), NC theory justifies the use of NCM-based classifiers to maintain feature-classifier alignment across incremental steps, inspiring methods that utilize fixed ETF structures or prototype alignment to integrate new categories with limited data while preserving old class knowledge (Yang et al., 2023; Tu et al., 2025b).

**Self-Organizing Incremental Neural Network (SOINN)** is a classic neural network designed for unsupervised incremental learning (Furao & Hasegawa, 2006). It represents the topology of non-stationary data streams by dynamically adjusting nodes and edges. To extend its capabilities to supervised tasks, several variants have been developed. For instance, the Enhanced SOINN adopts a single-layer structure with improved density-based noise removal (Furao et al., 2007). Furthermore, the Adjusted SOINN Classifier incorporates class labels into the incremental learning process (Shen & Hasegawa, 2008), enabling the network to handle online classification and adapt to overlapping class distributions.

## 3. Preliminary & Motivation

### 3.1. Problem Setting

Class-Incremental Learning (CIL) aims to learn from a stream of tasks $\mathcal{S} = \{\mathcal{T}_1, \mathcal{T}_2, \ldots, \mathcal{T}_T\}$ sequentially. For each task $\mathcal{T}_t$, the model is provided with a training dataset

$\mathcal{D}_t = \{(\mathbf{x}_i, y_i)\}_{i=1}^{n_t}$, where $\mathbf{x}_i \in \mathcal{X}$ represents an input sample and $y_i \in \mathcal{Y}_t$ is its corresponding label from the label space $\mathcal{Y}_t$. A core constraint in CIL is the disjoint nature of label spaces: $\mathcal{Y}_j \cap \mathcal{Y}_k = \emptyset$ for any $j \neq k$.

Upon transitioning to task $\mathcal{T}_t$, the datasets from previous tasks $\{\mathcal{D}_1, \ldots, \mathcal{D}_{t-1}\}$ are no longer accessible. The objective is to obtain a model that can correctly classify samples from the union of all observed classes $\mathcal{C}_t = \bigcup_{j=1}^{t} \mathcal{Y}_j$. During inference, given a test sample $\mathbf{x}$, the model must predict $\hat{y} \in \mathcal{C}_t$ without knowing the task ID, which is referred to as the class-incremental setting.

### 3.2. Pre-trained Models and the NCM Classifier Paradigm

In the context of modern CIL, the model typically consists of a powerful pre-trained feature extractor $f_\theta : \mathcal{X} \to \mathbb{R}^d$ and a classification mechanism. The NCM classifier is the prevailing choice for maintaining performance across incremental steps. For each class $c \in \mathcal{C}_t$, a prototype $\boldsymbol{\mu}_c$ is defined as the mean embedding of its training features:

$$\boldsymbol{\mu}_c = \frac{1}{|\mathcal{D}_c|} \sum_{(\mathbf{x},y)\in\mathcal{D}_c} f_\theta(\mathbf{x}), \qquad (1)$$

where $\mathcal{D}_c$ denotes the set of samples belonging to class $c$. During inference, the classification rule for a new sample $\mathbf{x}$ is defined by the maximum cosine similarity:

$$\hat{y} = \underset{c\in\mathcal{C}_t}{\arg\max} \cos\langle f_\theta(\mathbf{x}), \boldsymbol{\mu}_c \rangle = \underset{c\in\mathcal{C}_t}{\arg\max} \frac{f_\theta(\mathbf{x}) \cdot \boldsymbol{\mu}_c}{\|f_\theta(\mathbf{x})\|_2 \|\boldsymbol{\mu}_c\|_2}. \qquad (2)$$

### 3.3. Motivation: Procrustes Distance Experiment

To investigate feature drift, we conduct a geometric analysis on Split CIFAR-100 and Split CUB-200 by tracking the structural evolution of initial classes ($\mathcal{C}_{init}$). We quantify manifold deformation using the Average Procrustes Distance ($d_P^{(t)}$) between the initial (Task 1) and current (Task $t$) representations:

$$d_P^{(t)} = \frac{1}{|\mathcal{C}init|} \sum_{c\in\mathcal{C}_{init}} d_P(\mathbf{H}_c^{(1)}, \mathbf{H}_c^{(t)}), \qquad (3)$$

where $\mathbf{H}_c^{(1)}$ and $\mathbf{H}_c^{(t)}$ denote feature matrices for class $c$. We track representative PEFT methods (details in Section 5).

As shown in Figure 1, we use an empirical threshold of 0.1 for quasi-linear drift. On CIFAR-100, we observe that $d_P^{(t)}$ consistently exceeds this bound immediately from Task 2 and steadily climbs towards $0.35 - 0.40$. On CUB-200, methods like DualPrompt show a rapid increase in manifold distortion. This upward trend confirms that continuous

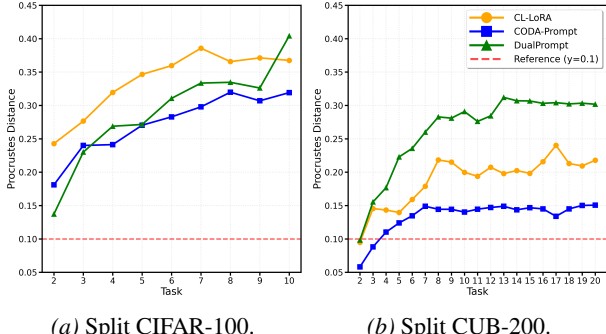

*(a)* Split CIFAR-100.      *(b)* Split CUB-200.

*Figure 1.* Evolution of the Average Procrustes Distance ($d_P^{(t)}$) on the initial classes across incremental tasks. The red dashed line ($y = 0.1$) indicates the empirical threshold for quasi-linear drift.

training causes the feature space to deviate fundamentally from the initial structure. Consequently, class distributions evolve into complex manifolds rather than maintaining the compact structure predicted by Neural Collapse, challenging the optimality of the single-prototype NCM assumption in practical CIL scenarios.

## 4. The Proposed Methods

### 4.1. HC-SOINN: Topological Manifold Representation

To accurately represent the class manifolds formed due to incomplete neural collapse, we propose the **HC-SOINN** (Hierarchical-Cluster Self-Organizing Incremental Neural Network) classifier. Unlike the standard NCM which assumes a unimodal Gaussian distribution (single prototype), HC-SOINN models each class $c$ as a topology graph $\mathcal{G}_c = (\mathcal{V}_c, \mathcal{E}_c)$, where $\mathcal{V}_c = \{\mathbf{v}_1, \ldots, \mathbf{v}_{K_c}\}$ represents a set of local sub-prototypes and $\mathcal{E}_c$ denotes the connectivity edges capturing the manifold's shape.

The construction of $\mathcal{G}_c$ proceeds in a coarse-to-fine manner, integrating the stability of hierarchical clustering with the adaptability of SOINN.

#### 4.1.1. HIERARCHICAL INITIALIZATION

Directly applying incremental clustering (like SOINN) on raw data streams is often sensitive to noise and input order. To mitigate this, we first accumulate feature samples in a memory buffer. Once the buffer is full or a task concludes, we perform Agglomerative Hierarchical Clustering to initialize the manifold skeleton (Johnson, 1967).

Given a set of normalized features $\mathbf{Z}_c = \{\tilde{f}_\theta(\mathbf{x}) \mid y = c\}$, we compute the linkage matrix using the **average linkage (UPGMA) criterion** combined with the cosine distance metric. Specifically, the distance between two clusters $\mathcal{C}_i$ and $\mathcal{C}_j$ is defined as the average cosine distance between all

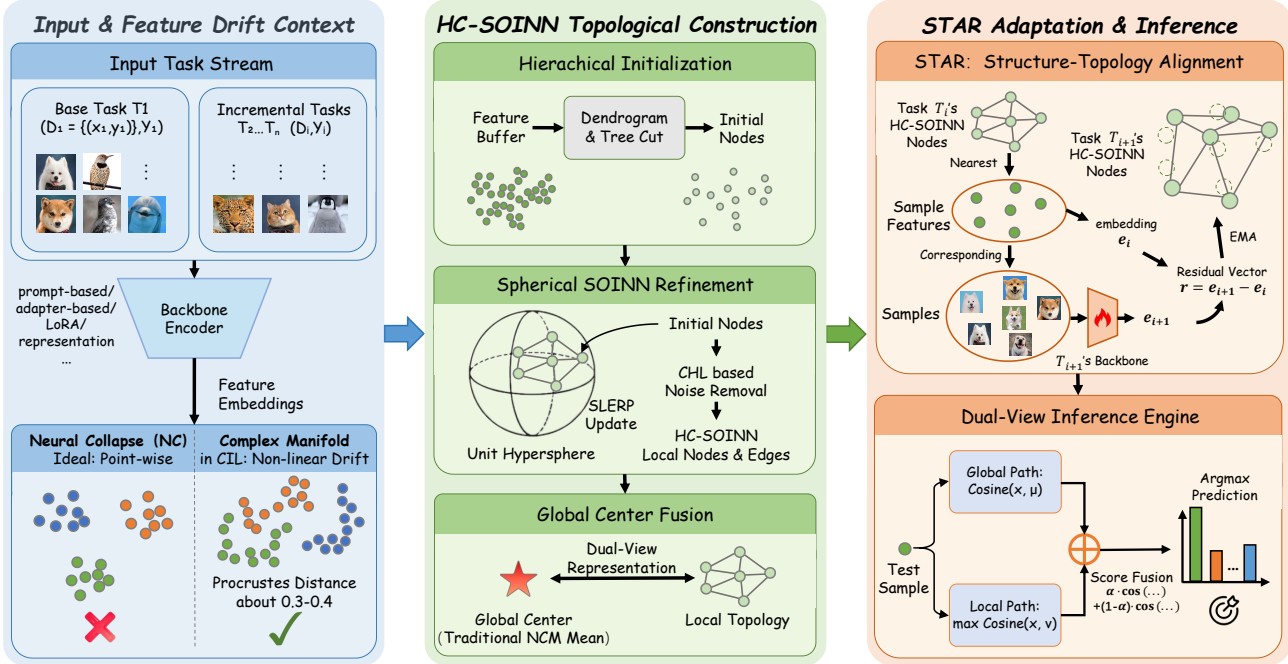

*Figure 2.* Overall pipeline of HC-SOINN and STAR. The method first constructs a topology-aware hierarchical classifier on fixed feature embeddings to model complex class manifolds that arise during class-incremental learning. It then employs the STAR mechanism to actively adapt this topology to non-linear feature drift via pointwise tracking, enabling dual-view inference without retraining.

inter-cluster pairs:

$$d(\mathcal{C}_i, \mathcal{C}_j) = \frac{1}{|\mathcal{C}_i||\mathcal{C}_j|} \sum_{\mathbf{u} \in \mathcal{C}_i} \sum_{\mathbf{v} \in \mathcal{C}_j} (1 - \cos\langle\mathbf{u}, \mathbf{v}\rangle). \quad (4)$$

The resulting dendrogram is cut to obtain a target number of clusters $K_{init}$, yielding an initial set of nodes $\mathcal{V}_c^{(0)}$. This specific linkage criterion is chosen for its robustness to outliers and its ability to produce compact, spherical clusters on the hypersphere, providing a stable initialization for the subsequent self-organizing phase.

### 4.1.2. SPHERICAL SOINN REFINEMENT

To capture the fine-grained geometry and connectivity of the manifold, we apply a simplified SOINN mechanism on the initial nodes $\mathcal{V}_c^{(0)}$. We adapt the standard Competitive Hebbian Learning (CHL) (Martinetz et al., 1991) to the unit hypersphere geometry imposed by the normalization.

For each iteration, given an input signal $\mathbf{z} \in \mathbf{Z}_c$ (or a cluster center from the previous phase), we identify the winner node $s_1$ and the runner-up $s_2$ from $\mathcal{V}_c$:

$$s_1 = \underset{\mathbf{v}_i \in \mathcal{V}_c}{\mathrm{argmax}} \cos\langle\mathbf{z}, \mathbf{v}_i\rangle, \quad s_2 = \underset{\mathbf{v}_i \in \mathcal{V}_c \setminus \{s_1\}}{\mathrm{argmax}} \cos\langle\mathbf{z}, \mathbf{v}_i\rangle.$$
$$(5)$$

**Topology Learning:** If $s_1$ and $s_2$ are not connected, an edge $(s_1, s_2)$ is created in $\mathcal{E}_c$. Each edge has an age attribute that increments over time; edges exceeding a maximum age

$age_{max}$ are removed. This mechanism dynamically learns the adjacency relations within the manifold.

**Spherical Update:** Standard SOINN updates node positions via linear interpolation. However, in our setting, nodes must remain on the unit hypersphere. Therefore, we employ **Spherical Linear Interpolation (SLERP)** (Shoemake, 1985) to update the winner $s_1$ and its topological neighbors $\mathcal{N}(s_1)$:

$$\mathbf{v}_{s_1} \leftarrow \mathrm{SLERP}(\mathbf{v}_{s_1}, \mathbf{z}, \eta_1),$$
$$\mathbf{v}_j \leftarrow \mathrm{SLERP}(\mathbf{v}_j, \mathbf{z}, \eta_2) \ \forall j \in \mathcal{N}(s_1), \quad (6)$$

where $\eta_1, \eta_2$ are learning rates. This ensures that the sub-prototypes evolve along the manifold surface rather than drifting into the interior of the hypersphere.

### 4.1.3. DUAL-VIEW INFERENCE

During inference, HC-SOINN combines a "Global View" provided by the class mean $\boldsymbol{\mu}_c$ (from NCM) and a "Local View" provided by the nearest sub-prototype in the learned topology. The decision score for a test sample $\mathbf{x}$ w.r.t class $c$ is defined as:

$$S(\mathbf{x}, c) = \alpha \cdot \cos\langle\tilde{f}_\theta(\mathbf{x}), \boldsymbol{\mu}_c\rangle + (1 - \alpha) \cdot \max_{\mathbf{v} \in \mathcal{V}_c} \cos\langle\tilde{f}_\theta(\mathbf{x}), \mathbf{v}\rangle,$$
$$(7)$$

where $\alpha \in [0, 1]$ balances the global robustness and local precision.

**Algorithm 1** HC-SOINN Construction and Inference

**Input**: Training stream $\mathcal{D}$, Feature Extractor $f_\theta$, Balance factor $\alpha$
**Output**: Class Topologies $\{\mathcal{G}_c\}_{c\in\mathcal{C}}$ and Global Means $\{\boldsymbol{\mu}_c\}_{c\in\mathcal{C}}$

1: **Training Phase:**
2: **for** each class $c$ in current task **do**
3:  Accumulate features $\mathbf{Z}_c = \{\tilde{f}_\theta(\mathbf{x})\}$
4:  Update global mean: $\boldsymbol{\mu}_c \leftarrow \text{Normalize}(\text{Mean}(\mathbf{Z}_c))$
5:  *// Phase 1: Hierarchical Initialization*
6:  $\mathcal{V}_c^{(0)} \leftarrow \text{HierarchicalClustering}(\mathbf{Z}_c, \text{metric = 'cosine'})$
7:  *// Phase 2: Spherical SOINN Refinement*
8:  Initialize $\mathcal{V}_c \leftarrow \mathcal{V}_c^{(0)}, \mathcal{E}_c \leftarrow \emptyset$
9:  **for** iter $= 1$ to $T_{soinn}$ **do**
10:   **for** $\mathbf{z} \in \mathbf{Z}_c$ **do**
11:    Find winner $s_1$ and runner-up $s_2$
12:    Create or refresh edge $(s_1, s_2)$ in $\mathcal{E}_c$
13:    Update $\mathbf{v}_{s_1}$ and neighbors via $\text{SLERP}(\cdot, \mathbf{z}, \eta)$
14:    Remove edges with age $> age_{max}$
15:   **end for**
16:   Remove isolated nodes from $\mathcal{V}_c$
17:  **end for**
18: **end for**
19: **Inference Phase:**
20: **for** test sample $\mathbf{x}$ **do**
21:  $\hat{y} = \underset{c}{\arg\max} \left( \alpha \tilde{f}_\theta(\mathbf{x})^\top \boldsymbol{\mu}_c + (1-\alpha) \underset{\mathbf{v}\in\mathcal{V}_c}{\max} \tilde{f}_\theta(\mathbf{x})^\top \mathbf{v} \right)$
22: **end for**

The complete training and inference procedure is summarized in Algorithm 1, and detailed implementation and parameter settings are provided in Section G.

The rationale behind this dual-view inference is to rectify the decision boundary. The global term $\cos\langle\tilde{f}_\theta(\mathbf{x}), \boldsymbol{\mu}_c\rangle$ acts as a stable baseline, ensuring the sample is generally aligned with the class centroid. The local term $\max_{\mathbf{v}\in\mathcal{V}_c} \cos\langle\tilde{f}_\theta(\mathbf{x}), \mathbf{v}\rangle$ serves as a topological refinement, awarding higher confidence if the sample falls into a specific high-density region (local manifold) of the class, even if it is far from the global mean. The class with the highest composite score is selected as the final prediction:

$$\hat{y} = \underset{c\in\mathcal{C}_t}{\arg\max} \, S(\mathbf{x}, c). \quad (8)$$

### 4.2. STAR: Structure-Topology Alignment via Residuals

To address the non-linear feature drift revealed in Section 3.3, we propose the **STAR** module. Unlike traditional methods that assume a global rigid transformation, STAR adopts a fine-grained Pointwise Trajectory Tracking strategy. It leverages the multi-node granularity of HC-SOINN to independently track and correct the drift of each topological

sub-prototype, allowing the manifold structure to actively deform and adapt to complex feature space evolutions.

For each learned class $c$, we establish a one-to-one mapping between its topological nodes $\mathcal{V}_c = \{\mathbf{v}_1, \ldots, \mathbf{v}_{K_c}\}$ and a set of anchor samples $\mathcal{A}_c = \{(\mathbf{x}_i, \mathbf{h}_i^{(ref)})\}_{i=1}^{K_c}$, where $\mathbf{x}_i$ is the representative sample for node $\mathbf{v}_i$. At the end of task $\mathcal{T}_t$, we compute the pointwise drift for each node by comparing the current feature $\mathbf{h}_i^{(t)} = f_{\theta_t}(\mathbf{x}_i)$ with the stored reference $\mathbf{h}_i^{(ref)}$.

To mitigate the instability caused by stochastic optimization steps, we employ an Exponential Moving Average (EMA) to smooth the drift trajectory. Let $\boldsymbol{\delta}_i$ denotes the tracked drift vector for the $i$-th node. The update rule is defined as:

$$\boldsymbol{\Delta}_i = \mathbf{h}_i^{(t)} - \mathbf{h}_i^{(ref)}, \quad \boldsymbol{\delta}_i \leftarrow (1-\lambda)\boldsymbol{\delta}_i + \lambda\boldsymbol{\Delta}_i, \quad (9)$$

where $\lambda \in (0, 1]$ is a momentum coefficient. This smoothed residual $\boldsymbol{\delta}_i$ is then applied to transport the corresponding unnormalized sub-prototype $\mathbf{v}_{raw,i}$ in the current feature space:

$$\mathbf{v}'_{raw,i} = \mathbf{v}_{raw,i} + \boldsymbol{\delta}_i. \quad (10)$$

After alignment, the nodes are re-normalized to the unit hypersphere. The class global mean $\boldsymbol{\mu}_c$ is synchronously updated using the average drift of all nodes: $\boldsymbol{\mu}'_c = \boldsymbol{\mu}_c + \frac{1}{K_c}\sum_i \boldsymbol{\delta}_i$. This mechanism ensures that the topology faithfully follows the non-linear migration of the feature distribution without expensive retraining. The complete algorithm of STAR is summarized in Algorithm 2 in Section C.

The overall pipeline of our framework is illustrated in Figure 2.

## 5. Performance Experiments

### 5.1. Experimental Setup

**Datasets and Metrics.** We evaluate our proposed method on three widely used benchmarks in Class-Incremental Learning (CIL): Split CIFAR-100 (generic objects), Split CUB-200 (fine-grained), and Split ImageNet-R (out-of-distribution robustness) (Krizhevsky et al., 2009; Wah et al., 2011; Hendrycks et al., 2021). For all datasets, we adhere to the standard class-incremental setting where task identities are not provided during inference. Detailed dataset statistics and splitting protocols are provided in Section G.

Following standard literature (Zhou et al., 2024a), we report two key metrics to evaluate performance:

**Average Accuracy** ($A_{Avg}$): The mean of classification accuracies obtained after each incremental task, defined as $A_{Avg} = \frac{1}{T}\sum_{t=1}^{T} \mathcal{A}_t$, where $\mathcal{A}_t$ is the test accuracy after task $t$.
**Last-Task Accuracy** ($A_{Last}$): The accuracy on all classes after the final task $T$ is completed, i.e., $A_{Last} = \mathcal{A}_T$.

*Table 1.* Performance comparison on Split CIFAR-100, Split CUB-200, and Split ImageNet-R. We report Average Accuracy ($A_{Avg}$) and Last-Task Accuracy ($A_{Last}$). "+ HC-SOINN" denotes our method. **Bold** indicates the best result in each column, and Red indicates the second-best result. The values in parentheses denote the performance gain/loss compared to the base method. All models adopt ViT-B/16-IN1K as the backbone.

| Benchmarks Setting | Split CIFAR-100 10 Tasks (10 classes/task) | | Split CUB-200 20 Tasks (10 classes/task) | | Split ImageNet-R 40 Tasks (5 classes/task) | |
|---|---|---|---|---|---|---|
| Methods | $A_{Avg}$ (%) | $A_{Last}$ (%) | $A_{Avg}$ (%) | $A_{Last}$ (%) | $A_{Avg}$ (%) | $A_{Last}$ (%) |
| SimpleCIL | 82.30 | 76.21 | 88.10 | 80.49 | 68.05 | 61.35 |
| w/ **HC-SOINN** | 83.91 (+1.61) | 78.29 (+2.08) | 91.37 (+3.27) | 85.96 (+5.47) | 69.97 (+1.92) | 63.93 (+2.58) |
| DualPrompt | 86.40 | 81.96 | 78.84 | 70.70 | 67.44 | 58.67 |
| w/ **HC-SOINN** | 88.95 (+2.55) | 82.98 (+1.02) | 83.72 (+4.88) | 77.31 (+6.61) | 73.79 (+6.35) | 69.27 (+10.60) |
| CODA-Prompt | 91.30 | 86.96 | 80.74 | 69.97 | 65.02 | 60.33 |
| w/ **HC-SOINN** | 91.62 (+0.32) | 87.56 (+0.60) | 89.43 (+8.69) | 85.75 (+15.78) | 72.23 (+7.21) | 70.67 (+10.34) |
| APER | 90.90 | 85.80 | 90.99 | 85.11 | 74.02 | 66.70 |
| w/ **HC-SOINN** | 92.07 (+1.17) | 87.39 (+1.59) | 91.52 (+0.53) | 86.39 (+1.28) | 76.06 (+2.04) | 69.33 (+2.63) |
| EASE | 91.92 | 87.25 | 88.18 | 81.21 | 79.30 | 72.08 |
| w/ **HC-SOINN** | 92.43 (+0.51) | 87.86 (+0.61) | 87.59 (-0.59) | 81.13 (-0.08) | 80.16 (+0.86) | 73.12 (+1.04) |
| SEMA | 91.17 | 86.04 | 79.52 | 65.56 | 73.48 | 66.63 |
| w/ **HC-SOINN** | 92.45 (+1.28) | 88.36 (+2.32) | **91.73** (+12.21) | **86.56** (+21.00) | 78.55 (+5.07) | 72.40 (+5.77) |
| CL-LoRA | 92.32 | 87.99 | 84.38 | 74.47 | 80.32 | 73.25 |
| w/ **HC-SOINN** | **92.62** (+0.30) | **88.52** (+0.53) | 84.46 (+0.08) | 74.60 (+0.13) | **80.67** (+0.35) | **73.60** (+0.35) |

## 5.2. Implementation Details

To demonstrate the universality of our approach, we integrated HC-SOINN into seven representative PTM-based CIL methods, covering diverse paradigms such as prompt tuning (DualPrompt, CODA-Prompt), adapter tuning (SEMA), LoRA (CL-LoRA), and representation learning (SimpleCIL, APER, EASE) (Wang et al., 2022b; Smith et al., 2023; Wang et al., 2025; He et al., 2025; Zhou et al., 2025; 2024b).

All experiments were conducted using the `LAMDA-PILOT` framework (Sun et al., 2025) with a standard ViT-B/16 backbone pre-trained on ImageNet-1K. We strictly follow the alignment principles of the framework to ensure fair comparisons. In our method, we simply replace the original classifier (e.g., FC layer or NCM) of the baseline with our HC-SOINN classifier, keeping the backbone training strategy unchanged. Specific hyper-parameter settings and detailed descriptions of each baseline are deferred to Section G. We maintain a single unified set of hyperparameters across all benchmarks to demonstrate the tuning-free robustness of our topological method (see Section H for sensitivity analysis).

## 5.3. Main Results

The quantitative comparisons on the three benchmarks are summarized in Table 1. It is evident that the integration

of HC-SOINN yields consistent and often substantial performance improvements across all three benchmarks and seven baseline architectures. It is particularly worth noting that methods such as CODA-Prompt and SEMA originally rely on parametric Fully Connected (FC) layers, which renders them susceptible to severe catastrophic forgetting in fine-grained (CUB-200) and long-sequence (ImageNet-R) tasks. The remarkable boosts observed (e.g., SEMA gains +21.00% on CUB-200) indicate that replacing the FC layer with our non-parametric, topology-aware classifier effectively mitigates this forgetting.

Moreover, HC-SOINN significantly revitalizes established methods, enabling them to rival or even surpass the latest SOTA baselines (SEMA and CL-LoRA). For instance, on Split CIFAR-100, the representation-based method EASE, when equipped with HC-SOINN, achieves the $A_{Last}$ of 87.86%, outperforming the standard SEMA (86.04%) and performing on par with CL-LoRA (87.99%). Similarly, on Split CUB-200, CODA-Prompt + HC-SOINN (85.75%) significantly surpasses the baseline performance of both SEMA and CL-LoRA.

Ultimately, by equipping SEMA and CL-LoRA (SOTA methods in CVPR 2025) with HC-SOINN, we successfully push the performance boundaries further, establishing new SOTA performance across the evaluated datasets.

*Table 2.* Classifier performance comparison on three benchmarks. We fix the backbone and prompt tuning mechanism (DualPrompt or CODA-Prompt) and only vary the classification head. "+ STAR" denotes the integration of our drift alignment module. **Bold** indicates the best result, and Red indicates the second-best result within each backbone setting. Values in parentheses denote the gain/loss compared to HC-SOINN without STAR. All models adopt ViT-B/16-IN1K as the backbone.

| Benchmarks Setting | Split CIFAR-100 10 Tasks (10 classes/task) | | Split CUB-200 20 Tasks (10 classes/task) | | Split ImageNet-R 40 Tasks (5 classes/task) | |
|---|---|---|---|---|---|---|
| **Classifier** | $A_{Avg}$ (%) | $A_{Last}$ (%) | $A_{Avg}$ (%) | $A_{Last}$ (%) | $A_{Avg}$ (%) | $A_{Last}$ (%) |
| *Backbone: DualPrompt* | | | | | | |
| Original (FC) | 86.40 | 81.96 | 78.84 | 70.70 | 67.44 | 58.67 |
| NCM | 87.47 | 81.19 | 83.51 | 76.80 | 72.09 | 67.50 |
| KAC | 86.30 | 81.81 | 83.55 | 80.87 | 69.68 | 63.85 |
| **HC-SOINN** | 88.95 | 82.98 | 83.72 | 77.31 | 73.79 | 69.27 |
| **HC-SOINN + STAR** | **89.49** (+0.54) | **85.63** (+2.65) | **86.78** (+3.06) | **85.84** (+8.53) | **73.93** (+0.14) | **69.75** (+0.48) |
| *Backbone: CODA-Prompt* | | | | | | |
| Original (FC) | 91.30 | 86.96 | 80.74 | 69.97 | 65.02 | 60.33 |
| NCM | 90.55 | 86.21 | 88.59 | 84.18 | 70.45 | 68.68 |
| KAC | 92.34 | 87.39 | 86.38 | 76.76 | 73.53 | 70.73 |
| **HC-SOINN** | 91.62 | 87.56 | 89.43 | 85.75 | 72.23 | 70.67 |
| **HC-SOINN + STAR** | **92.65** (+1.03) | **89.67** (+2.11) | **90.37** (+0.94) | **86.17** (+0.42) | **73.71** (+1.48) | **71.85** (+1.18) |

## 5.4. Classifier Performance Analysis

To isolate the classifier's contribution from representation learning, we employed DualPrompt and CODA-Prompt as fixed feature extractors and evaluated four classifier paradigms on the frozen features:

Original (FC): The standard linear classifier trained with the cross-entropy loss.
NCM: The Nearest Class Mean classifier, representing the single-prototype paradigm derived from NC theory.
KAC: The Kolmogorov-Arnold Classifier (Hu et al., 2025), a non-linear classifier proposed in CVPR 2025.
HC-SOINN (Ours): Our proposed topological classifier, evaluated both with and without the STAR alignment module.

Table 2 reveals three key insights. First, parametric FC classifiers suffer from catastrophic forgetting in long-sequence tasks, whereas prototype-based methods maintain superior stability. Second, while the non-linear KAC shows promising plasticity, it is unstable; in scenarios like Split CIFAR-100 with DualPrompt, it underperforms the FC baseline, and its generalization on out-of-distribution tasks is limited. Third, STAR fundamentally elevates robustness. By enabling the topological structure to actively track non-linear feature drift, HC-SOINN + STAR establishes a decisive lead. For instance, on Split ImageNet-R with DualPrompt, our framework achieves an $A_{Last}$ of 69.75%, surpassing KAC (63.85%) by 5.90%. This demonstrates that our *adaptive topological alignment* offers a significantly more reliable solution for complex incremental shifts than KAC's fixed non-linear boundaries.

## 6. Discussion & Analysis

### 6.1. Computational Efficiency Analysis

We analyze the computational costs on the Split CIFAR-100 benchmark (at the final Task 10), using CODA-Prompt as the baseline. We also include the KAC classifier for comparison. Table 3 summarizes the results.

**Training Efficiency.** Our framework demonstrates highly competitive training efficiency. While the KAC method incurs a noticeable computational burden, increasing training time by 10.24%, our full method (STAR) adds a minimal overhead of only $\approx 3.96\%$ (from 906.57s to 942.43s). This efficiency is achieved via a class-wise parallelization strategy, where the independent topological refinement for each category is executed concurrently to maximize CPU utilization.

**Inference Efficiency.** Remarkably, our method introduces virtually zero inference latency overhead (+0.01%). Although the model computes distances to multiple topological sub-prototypes to capture complex manifolds—a process that typically slows down inference—we effectively mitigate this bottleneck by employing GPU-accelerated matrix operations to vectorize distance computations. Consequently, our inference speed remains strictly on par with the baseline (26.84s), matching the efficiency of KAC (+0.00%) while providing a more robust topology.

**Decomposition of Inference Latency.** To further contextualize the inference efficiency, we decompose the total latency into backbone processing (i.e., feature extraction via the ViT backbone and prompt modules) and classifier

*Table 3.* Computational cost analysis of CODA-Prompt on Split CIFAR-100 (Task 10). We performed our experiments using an Intel Core i9-13900K CPU and one NVIDIA RTX 4090 GPU. All results are averaged over three independent runs. The percentages in parentheses denote the *overhead ratio*, calculated as (Ours − Baseline)/Baseline.

| Method | Training Time (s) | Inference Time (s) |
|---|---|---|
| CODA (+ FC) | 906.57±5.23 | 26.84±0.06 |
| + NCM | 930.81±0.28 (+2.67%) | 26.80±0.03 (-0.14%) |
| + KAC | 999.44±2.95 (+10.24%) | 26.84±0.09 (+0.00%) |
| + HC-SOINN | 933.52±0.41 (+2.97%) | 26.84±0.07 (+0.00%) |
| + STAR (Full) | 942.43±0.31 (+3.96%) | 26.84±0.06 (+0.01%) |

*Table 4.* Detailed decomposition of inference latency. The total inference time is broken down into backbone processing (including prompt mechanics) and classifier evaluation. The backbone consistently accounts for over 99.70% of the total computational cost across all methods.

| Method | Total Time (s) | Backbone | | Classifier | |
|---|---|---|---|---|---|
| | | Time (s) | Ratio | Time (s) | Ratio |
| CODA (+ FC) | 26.8361±0.0622 | 26.8126±0.0621 | 99.91% | 0.0235±0.0003 | 0.09% |
| + NCM | 26.7994±0.0292 | 26.7663±0.0292 | 99.88% | 0.0331±0.0003 | 0.12% |
| + KAC | 26.8372±0.0854 | 26.7944±0.0801 | 99.84% | 0.0428±0.0057 | 0.16% |
| + HC-SOINN | 26.8368±0.0703 | 26.7704±0.0671 | 99.75% | 0.0663±0.0032 | 0.25% |
| + STAR (Full) | 26.8386±0.0596 | 26.7699±0.0569 | 99.74% | 0.0686±0.0077 | 0.26% |

evaluation, as detailed in Table 4. The results reveal that the backbone network overwhelmingly dominates the computational cost, consistently accounting for over 99.7% of the total inference time across all methods. In contrast, the time spent on the classification head is marginally small, peaking at only 0.26% (0.0686s) for our full STAR method. This breakdown confirms that the overall inference bottleneck is intrinsically constrained by the heavy deep backbone architecture rather than the classifier. Consequently, the inference speed of the classification head has a negligible impact on the overall system latency, demonstrating that our topological refinement strategy achieves significant performance gains without incurring any practical latency penalties.

## 6.2. Additional Topological Visualization

To provide a comprehensive qualitative analysis of feature drift and the adaptation capability of our framework, we present an extended t-SNE visualization in Figure 3. We specifically track the structural evolution of the initial 10 classes (introduced in Task 1) across different stages of incremental learning.

**Initial Topological Fidelity.** As observed in Figure 3(a), at the end of the base session (Task 1), the proposed HC-SOINN classifier (represented by large circles) successfully captures the complex topology of the class manifolds. Unlike the single-prototype NCM (marked with '×'), which is limited to the geometric center, our topological nodes span the entire distribution of the query samples (small dots).

*Table 5.* Ablation study of proposed components on Split CUB-200. "HC" denotes Hierarchical Clustering. "SOINN" indicates SOINN-based topological refinement. "STAR" denotes the drift alignment module.

| Method | HC | SOINN | STAR | Avg (%) | Last (%) |
|---|---|---|---|---|---|
| Baseline (NCM) | - | - | - | 80.74 | 69.97 |
| Pure SOINN | - | ✓ | - | 86.71 | 82.65 |
| Pure HC | ✓ | - | - | 88.20 | 83.80 |
| HC + STAR | ✓ | - | ✓ | 89.39 | 85.07 |
| HC-SOINN | ✓ | ✓ | - | 89.43 | 85.75 |
| **Ours (Full)** | ✓ | ✓ | ✓ | **90.37** | **86.17** |

This validates the foundational premise of our method: a "local-to-global" graph representation is superior in modeling high-dimensional feature spaces.

**The Challenge of Feature Drift.** Figure 3(b) illustrates the severe impact of feature drift after five subsequent incremental steps. As the backbone updates for new tasks, the feature embeddings of the old classes shift non-linearly. The standard HC-SOINN nodes, lacking an active alignment mechanism, fail to keep pace with this evolution. A distinct spatial mismatch is visible, where the topological nodes lag behind the actual sample clusters, leaving numerous peripheral samples uncovered. This visualizes the "topological lag" phenomenon, explaining why static topological methods degrade in performance over time.

**Effectiveness of STAR Alignment.** In contrast, Figure 3(c) demonstrates the efficacy of the STAR module under the same drifted conditions. By leveraging the stored anchor drift to estimate pointwise trajectories, STAR actively deforms the topological structure. The aligned nodes in Figure 3(c) re-establish accurate coverage of the drifted manifolds, effectively matching the new distribution. This comparison provides compelling evidence that shifting from "drift resistance" to "drift adaptation" is crucial for maintaining long-term topological fidelity in Class-Incremental Learning.

## 6.3. Ablation Study

To rigorously verify the contribution of each component, we conduct a comprehensive ablation study on Split CUB-200 with CODA-Prompt. We evaluate combinations of Hierarchical Clustering (HC), SOINN refinement, and the STAR alignment module. The results are detailed in Table 5.

**Effectiveness of Topological Initialization and Refinement.** Compared to the single-prototype Baseline (NCM, 80.74%), both Pure SOINN (86.71%) and Pure HC (88.20%) achieve significant performance gains, confirming the validity of topological representations. notably, the combination of these two components (HC-SOINN) yields a further improvement to 89.43%, surpassing both individual

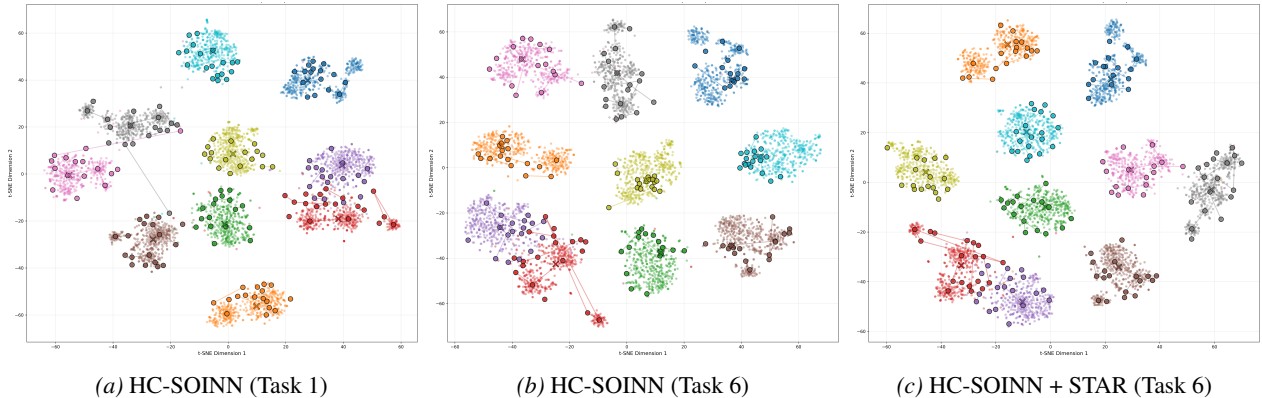

| *(a)* HC-SOINN (Task 1) | *(b)* HC-SOINN (Task 6) | *(c)* HC-SOINN + STAR (Task 6) |
|---|---|---|

*Figure 3.* t-SNE visualization of the feature distributions for the initial 10 classes. Small dots represent query samples, large circles denote HC-SOINN topological nodes, and '×' marks represent NCM prototypes. **(a)** At Task 1, HC-SOINN nodes perfectly capture the initial class manifolds. **(b)** By Task 6, without active alignment, the standard HC-SOINN nodes fail to cover the drifted features (highlighted by the spatial mismatch). **(c)** With STAR enabled, the topological nodes are actively deformed to match the drifted distribution at Task 6, restoring representational fidelity.

modules. This synergy arises from their complementary nature: Hierarchical Clustering provides a robust global initialization that mitigates SOINN's sensitivity to outliers during the early learning phase, while SOINN's incremental learning mechanism possesses a stronger capacity to represent fine-grained non-linear manifold features than static clustering.

**Orthogonality of STAR Alignment.** We evaluate the STAR module under different topological settings to verify its robustness. Adding STAR to Pure HC (HC+STAR) improves performance to $89.39\%$, proving that STAR effectively aligns distributions even with coarse cluster centroids. Furthermore, our full method (Ours), which applies STAR to the refined HC-SOINN topology, achieves the peak performance of $90.37\%$ ($A_{avg}$) and $86.17\%$ ($A_{last}$). The consistent gains observed in both settings demonstrate that Drift Adaptation (via STAR) and Topological Refinement (via HC-SOINN) are orthogonal and complementary mechanisms—one fixes the spatial misalignment, while the other optimizes the decision boundary structure.

## 7. Conclusion

In this paper, we revisit the theoretical optimality of the Nearest Class Mean (NCM) classifier within the context of Class-Incremental Learning. We identify that due to incomplete Neural Collapse, class representations in practical CIL scenarios manifest as complex manifolds rather than collapsed points, rendering single-prototype classifiers suboptimal. To bridge this gap, we propose **HC-SOINN**, a novel topology-aware classifier that captures the intrinsic geometry of class manifolds via a coarse-to-fine learning mechanism. Furthermore, we introduce **STAR**, a residual-based alignment module that shifts the paradigm from "drift

resistance" to "drift adaptation." By leveraging fine-grained pointwise trajectory tracking, STAR enables the topological structure to actively deform, precisely accommodating the non-linear feature evolution. Extensive experiments on three diverse benchmarks demonstrate that our topological framework consistently outperforms traditional NCM-based approaches. Future work will explore extending this topological representation to multi-modal continual learning scenarios to further validate its generalization capability. **Limitation.** We acknowledge that the performance gains come with a trade-off. Compared to the lightweight NCM, our framework introduces moderate computational overhead due to topological maintenance and requires additional storage for STAR's anchor buffer. Additionally, to maintain strict inference efficiency, our current scoring mechanism relies solely on node distances, leaving the rich edge connectivity unexploited during the classification phase.

## Acknowledgements

This work was partially supported by the National Natural Science Foundation of China (Grant Nos. 62495090, 62495094 and 62276127), Fundamental and Interdisciplinary Disciplines Breakthrough Plan of the Ministry of Education of China (No. JYB2025XDXM118), and the "111 Center" (No. B26023).

## Impact Statement

In this paper, we introduce HC-SOINN and STAR, a framework designed to address non-linear feature drift in Class-Incremental Learning (CIL). Broader Applications and Benefits: Our research advances the capability of AI systems to learn continuously from streaming data, which is critical for applications such as autonomous robotics, personalized

assistants, and dynamic medical diagnosis systems. By enabling models to adapt to new tasks without forgetting previous knowledge, our approach promotes the deployment of Lifelong Learning agents in real-world environments. Furthermore, compared to retraining models from scratch, our method significantly reduces computational costs and energy consumption over the model's lifecycle, contributing to the goal of Green AI and sustainable computing.

While improving stability, the deployment of dynamic topological models introduces specific challenges. Unlike static models, our system (driven by the STAR alignment module) actively evolves its decision boundaries. This dynamic nature complicates safety verification and interpretability, which is a concern in safety-critical domains like autonomous driving. If the pointwise trajectory tracking misaligns due to noisy data, it could lead to unpredictable failures. There is a risk that biases present in the initial training tasks could be encoded into the topological structure (HC-SOINN) and propagated or amplified in subsequent incremental steps. Dependence: As systems become better at adapting automatically, there is a risk of over-reliance on the model's self-correction capabilities, potentially reducing human oversight in monitoring data quality.

We encourage future research to explore methods for validating the stability of evolving topologies in real-time. Additionally, researchers should investigate mechanisms to detect and rectify biases within the learned manifold structures, ensuring that the "drift adaptation" process does not inadvertently entrench unfair decision patterns.

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

# A. Theoretical Analysis

## A.1. Proposition 1: Error Bound Analysis under Feature Drift

Here, we provide a theoretical bound to demonstrate that our topology-aware representation (HC-SOINN) minimizes the representation error under the "Feature Drift" problem compared to single-prototype NCM.

Let $\mathcal{M}_c \subset \mathbb{R}^d$ denote the feature manifold of class $c$ at task $t$. Let $\phi : \mathbb{R}^d \to \mathbb{R}^d$ be the drift function mapping features from task $t$ to $t + 1$ due to backbone updates. We formally define the *Representation Error* as the maximum deviation between a drifted sample and its corresponding drifted prototype representation.

**Assumption A.1** (Lipschitz Continuous Drift)**.** The feature drift function $\phi$ is $L$-Lipschitz continuous, meaning there exists a constant $L > 0$ such that for any $\mathbf{x}, \mathbf{y} \in \mathcal{M}_c$:

$$\|\phi(\mathbf{x}) - \phi(\mathbf{y})\| \leq L\|\mathbf{x} - \mathbf{y}\|. \tag{11}$$

This assumption is mild in deep learning, as gradient updates are typically bounded (e.g., by gradient clipping or weight decay), preventing arbitrary discontinuities in the feature mapping.

**Analysis of NCM Classifier.** The NCM classifier represents the manifold $\mathcal{M}_c$ using a single global centroid $\boldsymbol{\mu}_c$. The representation error for NCM after drift, denoted as $\mathcal{E}_{NCM}$, is bounded by the worst-case distance between a drifted sample and the drifted mean:

$$\mathcal{E}_{NCM} = \max_{\mathbf{z} \in \mathcal{M}_c} \|\phi(\mathbf{z}) - \phi(\boldsymbol{\mu}_c)\|$$
$$\leq \max_{\mathbf{z} \in \mathcal{M}_c} L\|\mathbf{z} - \boldsymbol{\mu}_c\| = L \cdot R_c, \tag{12}$$

where $R_c = \max_{\mathbf{z} \in \mathcal{M}_c} \|\mathbf{z} - \boldsymbol{\mu}_c\|$ represents the *Global Radius* of the class manifold. In non-collapsed scenarios, $R_c$ is significantly large.

**Analysis of HC-SOINN Classifier.** HC-SOINN partitions the manifold $\mathcal{M}_c$ into $K$ Voronoi regions $\{\mathcal{V}_1, \ldots, \mathcal{V}_K\}$ centered at sub-prototypes $\{\mathbf{v}_1, \ldots, \mathbf{v}_K\}$. A sample $\mathbf{z}$ is represented by its nearest sub-prototype $\mathbf{v}_{k^*}$. The representation error $\mathcal{E}_{HC}$, assuming the topology is preserved during drift, is bounded by:

$$\mathcal{E}_{HC} = \max_k \max_{\mathbf{z} \in \mathcal{V}_k} \|\phi(\mathbf{z}) - \phi(\mathbf{v}_k)\|$$
$$\leq \max_k \max_{\mathbf{z} \in \mathcal{V}_k} L\|\mathbf{z} - \mathbf{v}_k\| = L \cdot r_{local}, \tag{13}$$

where $r_{local} = \max_k \max_{\mathbf{z} \in \mathcal{V}_k} \|\mathbf{z} - \mathbf{v}_k\|$ is the maximum *Local Radius* of the sub-clusters.

**Conclusion & Impact on Forgetting.** Since hierarchical clustering and SOINN explicitly minimize quantization error, the manifold is decomposed into compact local regions. By definition of partitioning, $r_{local} \ll R_c$ holds for any complex manifold (e.g., non-convex shapes). Consequently:

$$\mathcal{E}_{HC} \leq L \cdot r_{local} \ll L \cdot R_c = \text{Upper Bound of } \mathcal{E}_{NCM}. \tag{14}$$

This inequality theoretically guarantees that HC-SOINN maintains a tighter error bound under feature drift. Even if the backbone distorts the space (scaling by $L$), the multi-prototype structure preserves local neighborhood relations better than a single rigid centroid. Crucially, this minimized representation error directly translates to reduced catastrophic forgetting. In class-incremental scenarios, forgetting occurs when feature drift causes severe nonlinear distortion of established decision boundaries. By tightly bounding the feature deviation within a much smaller local radius $r_{local}$, our topology-aware approach prevents drastic shifts in class decision half-spaces. Therefore, old-class samples remain correctly aligned with their corresponding sub-prototypes, directly preserving classification accuracy and mitigating forgetting without requiring rehearsal.

## A.2. Proposition 2: Bayesian Optimality via vMF Mixture

While Proposition 1 establishes robustness against drift, we further justify the specific form of our inference score $S(\mathbf{x}, c)$ (Eq. 6). Since feature vectors $\tilde{f}_\theta(\mathbf{x})$ are normalized to the unit hypersphere $\mathbb{S}^{d-1}$, the Gaussian assumption employed by standard NCM is geometrically suboptimal. Instead, we adopt the **von Mises-Fisher (vMF)** distribution, which is the maximum entropy distribution on the sphere.

**Definition A.2** (vMF Distribution). A random variable $\mathbf{x} \in \mathbb{S}^{d-1}$ follows a vMF distribution with mean direction $\boldsymbol{\mu} \in \mathbb{S}^{d-1}$ and concentration parameter $\kappa \geq 0$, denoted as $vMF(\mathbf{x}; \boldsymbol{\mu}, \kappa)$, if its probability density function is:

$$p(\mathbf{x}|\boldsymbol{\mu}, \kappa) = C_d(\kappa) \exp(\kappa \boldsymbol{\mu}^\top \mathbf{x}), \tag{15}$$

where $C_d(\kappa)$ is the normalization constant and $\boldsymbol{\mu}^\top \mathbf{x}$ corresponds to the cosine similarity.

To capture both the global stability (NCM) and local topological fidelity (HC-SOINN), we model the likelihood $P(\mathbf{x}|c)$ as a constrained hybrid distribution. Specifically, we assume the class conditional density is proportional to the product of a *Global Prior* (centered at $\boldsymbol{\mu}_c$) and a *Local Mixture* (centered at sub-prototypes $\mathcal{V}_c$):

$$\begin{aligned} P(\mathbf{x}|c) &\propto P_{global}(\mathbf{x}|c) \times P_{local}(\mathbf{x}|c) \\ &\approx vMF(\mathbf{x}; \boldsymbol{\mu}_c, \kappa_g) \times \max_{\mathbf{v} \in \mathcal{V}_c} vMF(\mathbf{x}; \mathbf{v}, \kappa_l), \end{aligned} \tag{16}$$

where $\kappa_g$ and $\kappa_l$ represent the concentration (confidence) of the global mean and local sub-prototypes, respectively. The $\max$ operator approximates the mixture sum, assuming locally disjoint high-density regions.

Under the assumption of uniform class priors $P(c)$, the Bayes Optimal Classifier maximizes the posterior $P(c|\mathbf{x})$, which is equivalent to maximizing the log-likelihood $\log P(\mathbf{x}|c)$. Substituting the vMF density into the log-likelihood, we obtain:

$$\begin{aligned} \log P(\mathbf{x}|c) &= \log \left( C_d(\kappa_g) e^{\kappa_g \boldsymbol{\mu}_c^\top \mathbf{x}} \cdot C_d(\kappa_l) e^{\kappa_l \max_{\mathbf{v}} \mathbf{v}^\top \mathbf{x}} \right) \\ &= \kappa_g \boldsymbol{\mu}_c^\top \mathbf{x} + \kappa_l \max_{\mathbf{v} \in \mathcal{V}_c} \mathbf{v}^\top \mathbf{x} + \underbrace{\log(C_d(\kappa_g) C_d(\kappa_l))}_{\text{class-independent constant}}. \end{aligned} \tag{17}$$

Since the feature vectors are $L_2$-normalized ($\|\mathbf{x}\| = \|\boldsymbol{\mu}\| = \|\mathbf{v}\| = 1$), the dot product is equivalent to the cosine similarity: $\mathbf{a}^\top \mathbf{b} = \cos\langle \mathbf{a}, \mathbf{b} \rangle$. Furthermore, assuming uniform concentration parameters across classes, the log-partition term is constant w.r.t. class $c$ and can be omitted during optimization.

Thus, the classification rule simplifies to maximizing the weighted sum of cosine similarities:

$$\hat{y} = \underset{c}{\arg\max} \left( \kappa_g \cos\langle \mathbf{x}, \boldsymbol{\mu}_c \rangle + \kappa_l \max_{\mathbf{v} \in \mathcal{V}_c} \cos\langle \mathbf{x}, \mathbf{v} \rangle \right). \tag{18}$$

To align this with our proposed dual-view metric, we exploit the scale-invariance property of the argmax operator. We define the total concentration $K_{total} = \kappa_g + \kappa_l$ and divide the objective by this positive constant without altering the prediction:

$$\begin{aligned} \hat{y} &= \underset{c}{\arg\max} \frac{1}{K_{total}} \left( \kappa_g \cos\langle \mathbf{x}, \boldsymbol{\mu}_c \rangle + \kappa_l \max_{\mathbf{v} \in \mathcal{V}_c} \cos\langle \mathbf{x}, \mathbf{v} \rangle \right) \\ &= \underset{c}{\arg\max} \left( \frac{\kappa_g}{K_{total}} \cos\langle \mathbf{x}, \boldsymbol{\mu}_c \rangle + \frac{\kappa_l}{K_{total}} \max_{\mathbf{v} \in \mathcal{V}_c} \cos\langle \mathbf{x}, \mathbf{v} \rangle \right). \end{aligned} \tag{19}$$

Finally, by letting the balancing factor $\alpha = \frac{\kappa_g}{\kappa_g + \kappa_l}$, which implies $1 - \alpha = \frac{\kappa_l}{\kappa_g + \kappa_l}$, we recover the inference formula:

$$\hat{y} = \underset{c}{\arg\max} \left( \alpha \cos\langle \mathbf{x}, \boldsymbol{\mu}_c \rangle + (1 - \alpha) \max_{\mathbf{v} \in \mathcal{V}_c} \cos\langle \mathbf{x}, \mathbf{v} \rangle \right). \tag{20}$$

**Conclusion.** This derivation proves that our heuristic score $S(\mathbf{x}, c)$ is theoretically equivalent to the Maximum A Posteriori (MAP) estimation under a hybrid vMF model. The hyperparameter $\alpha$ is not arbitrary; it explicitly represents the ratio of global confidence $\kappa_g$ to the total confidence. A higher $\alpha$ indicates that the global prototype is more reliable (e.g., in low-drift scenarios), while a lower $\alpha$ emphasizes local topological details.

## B. Implementation Insights and Geometric Rationale of HC-SOINN

In this section, we provide a more detailed exposition of the HC-SOINN classifier, focusing on the mathematical justification of its design and its advantages over traditional incremental clustering methods.

## B.1. Robustness of Hierarchical Initialization

As described in Section 4.1.1, we utilize Agglomerative Hierarchical Clustering with the **Unweighted Pair Group Method with Arithmetic Mean (UPGMA)** as the linkage criterion. The distance between two clusters $\mathcal{C}_i$ and $\mathcal{C}_j$ is calculated as:

$$l(\mathcal{C}_i, \mathcal{C}_j) = \frac{1}{|\mathcal{C}_i||\mathcal{C}_j|} \sum_{\mathbf{u} \in \mathcal{C}_i} \sum_{\mathbf{v} \in \mathcal{C}_j} d_{cos}(\mathbf{u}, \mathbf{v}), \tag{21}$$

where $d_{cos}(\mathbf{u}, \mathbf{v}) = 1 - \frac{\mathbf{u}^\top \mathbf{v}}{\|\mathbf{u}\|\|\mathbf{v}\|}$.

**Order-Independence and Noise Suppression:** Unlike standard SOINN or other online clustering methods that process signals one by one, HC-SOINN performs initialization on the entire feature buffer at the end of each task. This batch-style processing ensures the initialization is *order-independent*, mitigating the risk of the manifold skeleton being skewed by the input sequence. Furthermore, by averaging distances across all inter-cluster pairs, UPGMA effectively suppresses the influence of individual outliers, providing a stable and globally optimal "backbone" for the subsequent refinement.

## B.2. Spherical Geometry and SLERP Dynamics

A critical challenge in modern CIL is the utilization of normalized feature spaces (unit hyperspheres $\mathbb{S}^{d-1}$) to facilitate cosine similarity-based classification. Standard linear interpolation used in classic SOINN, defined as $\mathbf{v} \leftarrow \mathbf{v} + \eta(\mathbf{z} - \mathbf{v})$, would pull the node into the interior of the hypersphere, violating the unit-norm constraint and distorting the representation.

**SLERP for Consistency:** To maintain geometric integrity, we employ **Spherical Linear Interpolation (SLERP)**:

$$\text{SLERP}(\mathbf{v}, \mathbf{z}; \eta) = \frac{\sin((1-\eta)\Omega)}{\sin \Omega} \mathbf{v} + \frac{\sin(\eta\Omega)}{\sin \Omega} \mathbf{z}, \tag{22}$$

where $\Omega = \arccos(\mathbf{v}^\top \mathbf{z})$. SLERP ensures that sub-prototypes migrate strictly along the manifold surface. This guarantees that the distance metrics remain consistent throughout the training and inference phases, preventing representational collapse during the Competitive Hebbian Learning (CHL) phase.

## B.3. Topology as a Non-Linear Manifold Proxy

The graph structure $\mathcal{G}_c = (\mathcal{V}_c, \mathcal{E}_c)$ learned by HC-SOINN serves as a piecewise-linear approximation of the underlying class manifold.

- **Local Connectivity:** Connecting the winner and runner-up nodes (Top-2 nodes) essentially defines a Voronoi-like neighborhood structure, capturing the local density and shape of the data.

- **Co-evolution:** When a winner node $s_1$ and its topological neighbors $\mathcal{N}(s_1)$ are updated via SLERP, the entire local patch of the manifold "co-evolves" toward the new data distribution.

- **Refining Decision Boundaries:** In our Dual-View inference, the Local Path score $\max_{\mathbf{v} \in \mathcal{V}_c} \cos(\tilde{f}_\theta(\mathbf{x}), \mathbf{v})$ provides a fine-grained membership test. Unlike the single-prototype NCM which assumes a convex, unimodal distribution, the topology-aware grid can capture complex, non-convex shapes (e.g., "crescent" or "dumbbell" distributions). This allows the classifier to assign high confidence to samples that fall within the class's actual high-density regions, even if they are far from the global centroid $\boldsymbol{\mu}_c$.

# C. The Algorithm of STAR

The complete algorithm of STAR is summarized in Algorithm 2.

# D. Topological Complexity Analysis

A key property of our HC-SOINN classifier is its ability to adaptively determine the number of topological nodes required to represent each class, rather than using a fixed budget. To investigate the complexity of the manifolds learned by our framework, we report the average number of nodes per class across seven different backbones on three benchmarks. The results are summarized in Table 6.

---

**Algorithm 2** STAR Trajectory Alignment

---

**Input**: Old Classes $\mathcal{C}_{old}$, Anchors $\mathcal{A}$, Current Backbone $f_{\theta_t}$, HC-SOINN Classifier $\Psi$, Momentum $\lambda$
**Output**: Aligned Class Topologies

1: **for** each old class $c \in \mathcal{C}_{old}$ **do**
2:     $\boldsymbol{\Delta}_{list} \leftarrow []$
3:     **for** each node $i$ in $\Psi$.get_nodes($c$) **do**
4:         Retrieve anchor $(\mathbf{x}_i, \mathbf{h}_i^{(ref)})$ and stored drift $\boldsymbol{\delta}_i$
5:         Extract current feature: $\mathbf{h}_i^{(t)} \leftarrow f_{\theta_t}(\mathbf{x}_i)$
6:         Compute instant drift: $\boldsymbol{\Delta}_i \leftarrow \mathbf{h}_i^{(t)} - \mathbf{h}_i^{(ref)}$
7:         Update smoothed trajectory: $\boldsymbol{\delta}_i \leftarrow (1 - \lambda)\boldsymbol{\delta}_i + \lambda\boldsymbol{\Delta}_i$
8:         *// Pointwise Transport & Re-normalization*
9:         $\mathbf{v}'_{raw,i} \leftarrow \mathbf{v}_{raw,i} + \boldsymbol{\delta}_i$
10:        $\mathbf{v}'_i \leftarrow \mathbf{v}'_{raw,i}/\|\mathbf{v}'_{raw,i}\|$
11:        $\Psi$.update_node($c, i, \mathbf{v}'_{raw,i}, \mathbf{v}'_i$)
12:        Update Reference: $\mathbf{h}_i^{(ref)} \leftarrow \mathbf{h}_i^{(t)}$
13:        Append $\boldsymbol{\delta}_i$ to $\boldsymbol{\Delta}_{list}$
14:     **end for**
15:     *// Synchronize Global Mean*
16:     $\boldsymbol{\mu}'_c \leftarrow \boldsymbol{\mu}_c + \text{Mean}(\boldsymbol{\Delta}_{list})$
17:     $\Psi$.update_global_mean($c, \text{Normalize}(\boldsymbol{\mu}'_c)$)
18: **end for**

---

**Adaptive yet Compact Representation.** As shown in Table 6, the node count exhibits remarkable stability across different feature extractors (e.g., SimpleCIL vs. CODA-Prompt), consistently converging to a compact range. For standard datasets like Split CIFAR-100 and CUB-200, the topology stabilizes at approximately $18 \sim 19$ nodes per class. For Split ImageNet-R, which contains significant intra-class variance due to diverse artistic styles, the model adaptively allocates slightly more resources ($\approx 21$ nodes).

**Efficiency of Manifold Approximation.** These statistics demonstrate the efficiency of our "local-to-global" learning mechanism. By utilizing an average of roughly 20 sub-prototypes, HC-SOINN successfully captures the complex non-linear geometry of class manifolds. This number represents a "sweet spot"—it is significantly richer than the single prototype of NCM, yet remains sparse enough to avoid the high computational and storage burdens associated with storing all training samples.

*Table 6.* Average number of topological nodes per class generated by HC-SOINN across different methods and datasets.

| Base Method | Split CIFAR-100 | Split CUB-200 | Split ImageNet-R |
|---|---|---|---|
| SimpleCIL | 18.67 | 18.74 | 20.98 |
| DualPrompt | 18.98 | 19.04 | 21.38 |
| CODA-Prompt | 18.64 | 18.86 | 21.12 |
| APER | 17.24 | 17.54 | 18.93 |
| EASE | 19.66 | 19.28 | 21.80 |
| SEMA | 18.87 | 18.55 | 18.71 |
| CL-LoRA | 18.70 | 18.29 | 21.38 |
| **Average** | **18.68** | **18.61** | **20.61** |

*Table 7.* Comparison with rehearsal-based methods using the ViT-B/16-IN21K backbone. "Exemplars" denotes the average number of stored images per class. Our method (SEMA + Ours) achieves state-of-the-art performance with a storage budget comparable to or lower than the standard 20-image fixed buffer. The results for the baselines are sourced from (Zhou et al., 2024b).

| Benchmarks Setting | Split ImageNet-R 10 Tasks (20 classes/task) | | | Split CIFAR-100 10 Tasks (10 classes/task) | | |
|---|---|---|---|---|---|---|
| **Method** | **Exemplars** | $A_{Avg}$ (%) | $A_{Last}$ (%) | **Exemplars** | $A_{Avg}$ (%) | $A_{Last}$ (%) |
| iCaRL | 20 | 72.42 | 60.67 | 20 | 82.46 | 73.87 |
| DER | 20 | 80.48 | 74.32 | 20 | 86.04 | 77.93 |
| FOSTER | 20 | 81.34 | 74.48 | 20 | 89.87 | 84.91 |
| MEMO | 20 | 74.80 | 66.62 | 20 | 84.08 | 75.79 |
| SEMA (Base) | **0** | 80.51 | 73.83 | **0** | 92.56 | 88.16 |
| SEMA + Ours | 18.87 | **81.70**(+1.19) | **75.95**(+2.12) | 18.48 | **94.25**(+1.69) | **90.58**(+2.42) |

## E. Comparison with Traditional Exemplar-based Methods

To strictly evaluate the effectiveness of our exemplar usage, we benchmark our framework against classic rehearsal-based CIL methods, including iCaRL (Rebuffi et al., 2017), DER (Yan et al., 2021), FOSTER (Wang et al., 2022a), and MEMO (Zhou et al., 2022). We strictly follow the task settings in (Zhou et al., 2024b). We employ the ViT-B/16-IN21K backbone for all methods to ensure a fair comparison, integrating our HC-SOINN and STAR modules into the SEMA (Wang et al., 2025) baseline. The results on Split ImageNet-R (10 tasks, 20 classes each) and Split CIFAR-100 (10 tasks, 10 classes each) are summarized in Table 7.

As evidenced by the results, integrating our method into SEMA consistently outperforms all compared rehearsal-based approaches. On Split CIFAR-100, our method achieves an Average Accuracy of $94.25\%$, significantly surpassing the strongest rehearsal baseline, FOSTER ($89.87\%$). Similarly, on the challenging Split ImageNet-R, we maintain a performance edge ($81.70\%$ vs. $81.34\%$). Crucially, this superior performance is achieved with a lower storage budget. While traditional methods rely on a fixed buffer of 20 exemplars per class, our adaptive topology generates an average of only $18.48$ and $18.87$ nodes per class for CIFAR-100 and ImageNet-R, respectively. Furthermore, a fundamental distinction lies in the utilization of stored samples: while rehearsal methods require computationally expensive gradient-based retraining to alleviate forgetting, STAR utilizes these samples exclusively for inference-time spatial alignment without backward propagation. This implies that our drift adaptation mechanism is theoretically orthogonal to gradient-based replay. Future work will explore combining STAR's alignment with replay buffers to achieve a more exhaustive utilization of stored exemplars, potentially unlocking further performance gains.

## F. Integrating Original Classification Head Information

In our main experiments, the original parameterized classification head (i.e., the Fully Connected or FC layer) is replaced by our topology-aware HC-SOINN classifier. This design choice is primarily motivated by the vulnerability of standard FC layers to severe task-recency bias under Class-Incremental Learning (CIL) scenarios. However, it is natural to question whether the original head still retains complementary discriminative information that could be leveraged.

To explore the boundary of integrating these two paradigms, we introduce a dynamic score fusion mechanism during inference:

$$\text{Final Score} = (1 - w) \times \mathcal{S}_{\text{HC-SOINN}} + w \times \mathcal{S}_{\text{Calibrated FC}}, \tag{23}$$

where $w \in [0, 1]$ is the weight assigned to the original classification head. We evaluate this strategy using the DualPrompt + HC-SOINN backbone across three benchmarks, setting $w \in \{0, 0.3, 0.5\}$. The results are summarized in Table 8.

The empirical results reveal a clear trade-off:

- **Moderate fusion is beneficial** ($w = 0.3$): Assigning a conservative weight to the FC layer yields slight performance improvements on most datasets (e.g., $A_{last}$ on Split CIFAR-100 increases from $85.63\%$ to $86.14\%$). This confirms that the calibrated FC layer still contains supplementary knowledge that can complement our topological geometric

*Table 8.* Performance comparison of dynamic score fusion between HC-SOINN and the original FC classifier. $w = 0$ indicates our default setting without FC fusion.

| Dataset | $w = 0$ (No Fusion) | | $w = 0.3$ (Moderate Fusion) | | $w = 0.5$ (Over-reliance) | |
|---|---|---|---|---|---|---|
| | $A_{avg}$ (%) | $A_{last}$ (%) | $A_{avg}$ (%) | $A_{last}$ (%) | $A_{avg}$ (%) | $A_{last}$ (%) |
| Split CIFAR-100 | 89.49 | 85.63 | **89.68** | **86.14** | 89.54 | 85.59 |
| Split CUB-200 | 86.78 | **85.84** | **86.81** | **85.84** | 86.47 | 85.24 |
| Split ImageNet-R | **73.93** | 69.75 | 73.75 | **69.83** | 73.75 | 69.70 |

features.

- **Over-reliance is detrimental** ($w = 0.5$)**:** When the influence of the FC layer is further increased, overall accuracy noticeably declines (e.g., $A_{last}$ on Split CUB-200 drops to $85.24\%$, underperforming the baseline). This validates our initial concern: excessive participation of the FC layer reintroduces catastrophic forgetting and task bias, thereby overshadowing the inherent anti-forgetting advantages of the topology-aware architecture.

In conclusion, while a modest integration of the FC layer's knowledge can be slightly advantageous, the optimal weight is highly sensitive. The pure HC-SOINN representation ($w = 0$) remains a highly robust, simpler, and bias-free default for continuous evaluation.

## G. Detailed Implementation and Parameter Settings

### G.1. Datasets and Benchmarks

We evaluate our framework on three widely adopted Class-Incremental Learning (CIL) benchmarks: **Split CIFAR-100**, **Split CUB-200**, and **Split ImageNet-R**. These benchmarks span diverse visual recognition scenarios, including coarse-grained natural images, fine-grained categorization, and domain-shifted artistic renditions, enabling a comprehensive evaluation of incremental learning performance.

All experiments follow the standard CIL protocol, where classes are introduced sequentially in a series of disjoint tasks. During training, the model only has access to data from the current task, while during evaluation it is required to classify samples from all classes encountered so far.

**Split CIFAR-100**: Contains 100 natural image classes with a total of 60,000 images at a resolution of $32 \times 32$. Following common practice, the dataset is divided into 10 incremental tasks, each consisting of 10 classes. This benchmark is widely used as a standard testbed for evaluating class-incremental learning methods under limited-resolution settings.

**Split CUB-200**: A fine-grained visual classification dataset consisting of 200 bird species and 11,788 images. We split the dataset into 10 tasks, each introducing 20 new classes. Due to high inter-class similarity and subtle discriminative cues, Split CUB-200 poses significant challenges for maintaining discriminative representations in class-incremental learning.

**Split ImageNet-R**: Contains 200 classes with approximately 30,000 images, where images are artistic renditions such as cartoons, sketches, graffiti, and paintings. The dataset is divided into 40 tasks with 5 classes per task. The severe domain shift from natural images makes ImageNet-R particularly suitable for evaluating the robustness and generalization ability of incremental learning methods based on pre-trained models.

### G.2. Baselines

We compare our method with a comprehensive set of state-of-the-art class-incremental learning baselines. These methods can be broadly categorized into representation-based approaches and parameter-efficient fine-tuning (PEFT) methods. The latter further include prompt-based and adapter-based techniques, which are especially effective for transformer-based pre-trained models.

**SimpleCIL** is a strong baseline that freezes the pre-trained backbone and incrementally updates the classifier using class prototypes, demonstrating that well-generalized pre-trained representations alone can achieve competitive performance in class-incremental learning.

**DualPrompt** introduces both global prompts shared across tasks and task-specific prompts dedicated to individual tasks, achieving a balance between knowledge sharing and task discrimination in class-incremental learning.

**CODA-Prompt** further enhances prompt learning by dynamically composing and selecting prompts using attention mechanisms, improving prompt diversity and reducing task interference in long task sequences.

**APER** argues that class-incremental learning can be effectively addressed by combining highly generalizable pre-trained features with adaptive classifier updates. APER avoids rehearsal and parameter expansion, relying instead on adaptive prototype estimation to balance stability and plasticity across incremental tasks.

**EASE** addresses representation drift and class interference by incrementally expanding task-specific subspaces. It constructs an ensemble of expandable feature subspaces to better accommodate newly introduced classes while preserving previously learned knowledge.

**SEMA** proposes a mixture-of-adapters framework that allows the model to dynamically expand its capacity. By selectively activating and combining multiple adapters, SEMA enhances model expressiveness while mitigating catastrophic forgetting.

**CL-LoRA** adopts low-rank adaptation modules to incrementally fine-tune pre-trained models. It enables rehearsal-free class-incremental learning by introducing task-wise low-rank updates while keeping the backbone parameters frozen.

We also compare some exemplar-based methods in the main paper as follows:

**iCaRL** combines a nearest-class-mean classifier with a herding-based exemplar selection strategy. It utilizes knowledge distillation to mitigate catastrophic forgetting while learning new classes.

**DER** employs a dynamic architecture that expands the feature extractor for each new task. It integrates features from both frozen past extractors and the current learnable extractor to preserve old knowledge while acquiring new concepts.

**FOSTER** adopts a two-stage learning paradigm involving feature boosting and compression. It first dynamically expands the model to fit new task residuals and subsequently compresses the expanded parameters via distillation to maintain model compactness.

**MEMO** serves as a rehearsal-based baseline that utilizes stored exemplars to maintain the decision boundaries of previous classes, balancing the stability-plasticity trade-off through memory-efficient optimization.

### G.3. Implementation Details

**Backbone and Training.** Following standard protocols, we employ a Vision Transformer (**ViT-B/16**) pre-trained on ImageNet-1K (IN1K) as the feature extractor for all experiments, except as described in Section E. The backbone parameters remain frozen during the incremental stages to prevent catastrophic forgetting, while only the PEFT modules (if applicable) and the classifiers are updated.

**HC-SOINN Hyperparameters.** Our topological classifier, HC-SOINN, is configured with the following hyperparameters to ensure a balance between stability and plasticity:

**Balance Factor** ($\alpha$)**:** Set to 0.5. This equal weighting controls the contribution of the global class center versus the local sub-prototypes during inference (Eq. 7).

**Target Cluster Count** ($K_{init}$)**:** Set to 60. This parameter determines the granularity of the initial hierarchical clustering.

**Max Edge Age** ($age_{max}$)**:** Set to 20. Edges in the SOINN graph that have not been refreshed for 20 iterations are removed to prune outdated topological connections.

**SOINN Iterations** ($T_{soinn}$)**:** Set to 1. We perform a single refinement pass per task to maintain computational efficiency.

**STAR Momentum** ($\lambda$)**:** Set to 0.999. This coefficient controls the EMA when updating the pointwise drift vectors, smoothing the trajectory to mitigate instability caused by stochastic optimization steps during topological alignment.

Most experiments are conducted using the same hyperparameter settings detailed above. Our method achieves strong performance across all evaluations under this single set of hyperparameters, demonstrating its robustness.

All experiments are conducted on NVIDIA GPUs using PyTorch. For the baseline methods, we utilize their official implementations and recommended settings to ensure a fair comparison.

## H. Comprehensive Hyperparameter Sensitivity and Robustness

To comprehensively evaluate the model's sensitivity to the hyperparameters, we conducted extensive analyses on Split CIFAR-100 (integrated with CODA-Prompt) and Split ImageNet-R (integrated with SimpleCIL and DualPrompt). The detailed results are illustrated in Figure 4 and Figure 5.

**Detailed Parameter Analysis.** As clearly observed from the extensive evaluations, our framework maintains a high degree of performance stability across an extremely broad range of parameter values:

- **Balance Factor ($\alpha$):** As shown in Figure 4(a) and Figure 5(a), $\alpha$ reveals an "inverted-U" performance trend. Both extremes—pure NCM ($\alpha = 1.0$, $90.55\%$ on CIFAR-100) and pure local search ($\alpha = 0.0$, $89.34\%$)—are suboptimal. The balanced setting ($\alpha = 0.5$) consistently yields the peak accuracy ($91.62\%$), confirming the necessity of combining global stability with local plasticity.

- **Target Cluster Count ($K_{init}$):** While increasing nodes from 20 to 500 improves accuracy ($90.92\%$ to $92.28\%$ on CIFAR-100) by refining the manifold approximation (Figure 4b), the marginal utility eventually diminishes. We adopt $K_{init} = 60$ as a highly robust "sweet spot" that achieves excellent accuracy while maintaining strict computational and storage efficiency.

- **Topological Maintenance & Alignment ($age_{max}$, $T_{soinn}$, $\lambda$):** For the remaining parameters governing topological graph updates and pointwise drift tracking, the performance curves are remarkably flat. For instance, the framework shows extreme fault tolerance across a wide range of maximum edge ages ($age_{max} \in [3, 30]$) and STAR momentum values ($\lambda \in [0.9, 1.0]$).

**The "Tuning-Free" Philosophy.** Notably, dataset-specific fine-tuning could yield even higher metrics than those reported in our main results. For example, on ImageNet-R (Figure 5a), adjusting $\alpha$ to 0.6 achieves an $A_{Avg}$ of $70.43\%$, which noticeably surpasses our reported $69.97\%$ obtained under the unified $\alpha = 0.5$ setting. However, we deliberately bypassed such dataset-specific tuning. By reporting results using a single, unified configuration across all datasets, we demonstrate the "tuning-free" nature of our method. HC-SOINN inherently relies on data-driven self-organizing growth to dynamically depict complex manifolds, granting it remarkable adaptability and eliminating the practical burden of manual parameter engineering.

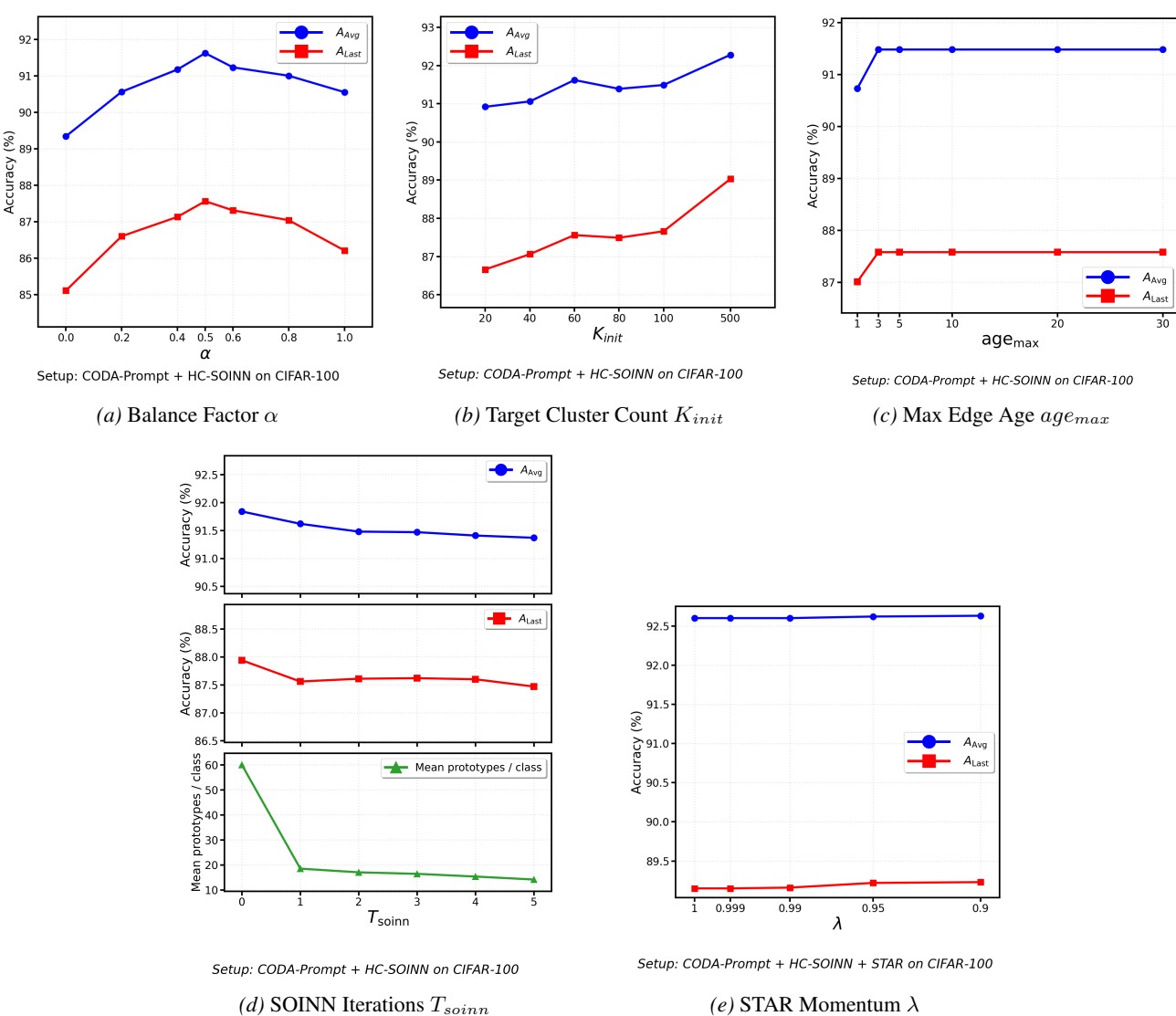

*(a)* Balance Factor $\alpha$      *(b)* Target Cluster Count $K_{init}$      *(c)* Max Edge Age $age_{max}$

*(d)* SOINN Iterations $T_{soinn}$      *(e)* STAR Momentum $\lambda$

*Figure 4.* Comprehensive hyperparameter sensitivity analysis on **Split CIFAR-100** (integrated with CODA-Prompt). The results demonstrate that our framework maintains highly stable average accuracy ($A_{Avg}$) and last-task accuracy ($A_{Last}$) across a wide range of values.

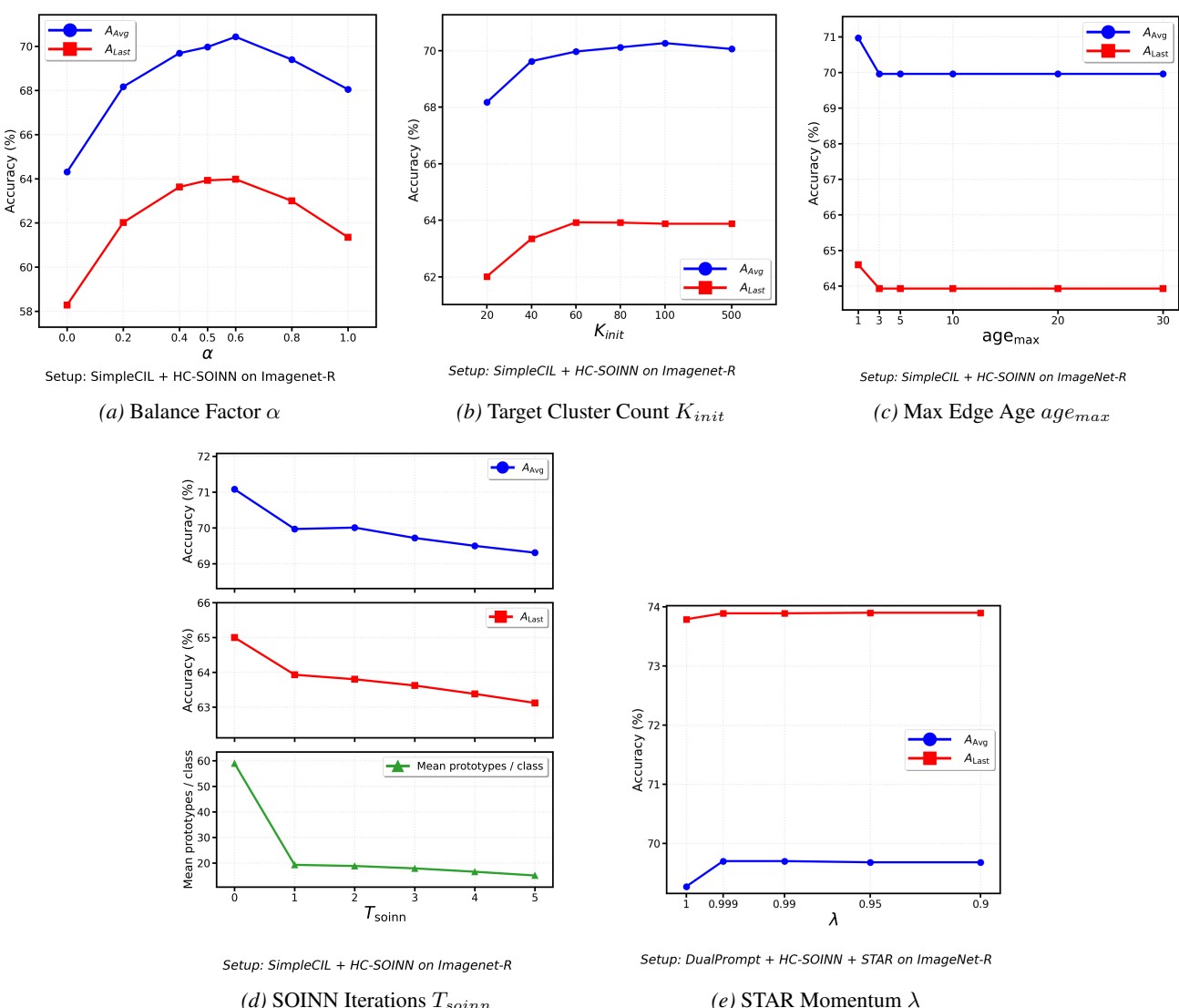

*(a)* Balance Factor $\alpha$

*(b)* Target Cluster Count $K_{init}$

*(c)* Max Edge Age $age_{max}$

*(d)* SOINN Iterations $T_{soinn}$

*(e)* STAR Momentum $\lambda$

*Figure 5.* Comprehensive hyperparameter sensitivity analysis on **Split ImageNet-R**. Parameters (a)-(d) are evaluated with SimpleCIL, while (e) is evaluated with DualPrompt. Similar to CIFAR-100, the performance remains remarkably robust across diverse configurations.

