# OpenReview forum: "Beyond Point-wise Neural Collapse: A Topology-Aware Hierarchical Classifier for Class-Incremental Learning"
_ICML.cc/2026/Conference — ICML 2026 regular_

### Official Review · Reviewer_Dy6q · 2026-03-06

**Soundness:** 3
**Presentation:** 2
**Significance:** 3
**Originality:** 2
**Overall Recommendation:** 5
**Confidence:** 3

**Summary:**

This paper proposed HC-SOINN for class-incremental learning (CIL). In particular, comparing to single-prototype NCM that uses class mean for classification, the proposed method learns a graph with certain topological structure for each class as prototypes, and uses the learned graph as well as the class mean for the classification. Further, the authors proposed STARThe authors also applied the proposed methods to on standard benchmarks, and compare to existent methods.

**Compliance With Llm Reviewing Policy:**

Affirmed.

**Final Justification:**

In the original review, the main concerns are on the intuition behind each part of the proposed methods, and the practical efficiency. In the rebuttal, the authors addressed these two point by discussing the intuitive meaning of each parts of the methods. I suggest the authors to include these points in the revision to make the paper more clear.

Overall, I think this paper proposed a new method for Class-Incremental Learning, which is well-motivated and seems to work well in practical experiments, thus I'm welling to increase my score to 5.

**Key Questions For Authors:**

1. When using the HC-SOINN for classification, only the information of the nodes are use,  and seems that the graph topology is only used to update the nodes. I wonder if it is possible to utilize the information of the whole graph topology for classification.

2. When integrate the HC-SOINN with other methods, the original classification head (such as FC layers) is replaced by HC-SOINN and not used. I wonder if it is possible to use the information of the original classification head in HC-SOINN?

**Limitations:**

Yes.

**Strengths And Weaknesses:**

### Strengths

1. This paper aim to improve the limitation of single-prototype classification due to potential feature dritft, I believe this is a well-motivated and interesting.

2. The proposed HC-SOINN + STAR methods seems to work well on standard benchmarks, given the experimental results in table 1 and table 2. Also the methods is easy to integrate into current CIL methods since one can simply replace the classifier layer. Thus I believe the proposed methods is solid and potentially useful from an experimental point of view.

### Weaknesses

1. While I understand the procedure of the proposed methods in general, I don't understand why each parts of the proposed methods are necessary and can potentially improve the classification performance intuitively. In particular:

    (1)  For the hierarchical initialization, the proposed methods use Agglomerative Hierarchical Clustering.Could the authors describe the methods briefly somewhere in the paper for completeness, and also briefly discuss why this method is more robust and can providing a more stable initialization? Also, could the authors also briefly explain how the robustness of the initial features affect later refinement and classification procedure, since anyway the nodes are updated by the signals.

    (2) For the spherical SOINN refinement, again, could the authors briefly describe SLERP method somewhere in the paper for completeness? Also, could the authors also explain intuitive why we connect the two nodes with top-2 alignment with the new given feature, and then why should we update the nodes by SLERP method? In addition, how the graph that constrcucted in this way can represent the topological structure of the features in the class, and can be useful for classification?

2. Another concern is the practical efficiency of the proposed. First of all, as the authors, as discussed by the authors in section 6, the inference time is increased compare to baseline methods (which is expected as a tradeoff for a better performance). But more importantly, it seems that there's a lot of parameters in the proposed methods, such as $\alpha, \eta_1, \eta_2, K_{init}, \lambda, age_{max}$ that might potentially affect the performace as partially discussed in section 6.3 of the paper, thus it might computationally inefficient to tune the hyperparameters given a new task.

---

> ### Author Rebuttal · Authors · 2026-03-31
>
> Dear Reviewer Dy6q,
>
> **(Note: All referenced figures/tables are at our anonymous link for rebuttal: [https://anonymous.4open.science/r/ICML2026-Rebuttal-D2D6/](https://anonymous.4open.science/r/ICML2026-Rebuttal-D2D6/).)**
>
> Thank you for your constructive evaluation and for recognizing our work. We address all your questions below:
>
> **1. Details About Hierarchical Initialization**
> We apologize for omitting these details due to space constraints and will add them to the appendix.
> * **Method & Robustness:** We use Agglomerative Hierarchical Clustering (HC) with the UPGMA (average linkage) criterion and cosine distance. Unlike purely online incremental clustering (original SOINN), which is highly sensitive to input order, HC performs a global bottom-up aggregation at the end of a task. This is order-independent and suppresses outliers, yielding a globally optimal and highly stable manifold "skeleton".
> * **Impact on Subsequent Steps:** SOINN is highly sensitive to initialization. HC's robust initialization ensures that subsequent SLERP updates are securely anchored to genuine high-density regions, avoiding wasted computations on false nodes and significantly enhancing topological stability and classification accuracy.
>
> **2. Inner Mechanisms of Spherical SOINN and SLERP**
> These intuitive physical explanations will be added to the revision.
> * **Necessity of SLERP:** Standard SOINN uses linear interpolation, which pulls nodes into the interior of our unit hypersphere feature space, breaking the cosine metric. SLERP strictly constrains nodes to move along the manifold surface, ensuring geometric consistency.
> * **Connecting Top-2 Nodes:** Connecting the two nodes closest to the input signal (Winner and Runner-up) establishes "local spatial connectivity," dynamically outlining high-density regions. Updating them via SLERP allows the entire local high-density region to "co-evolve".
> * **Graph Aiding Classification:** This graph tightly fits complex, non-convex manifolds, avoiding the linear "one-size-fits-all" division of single prototypes. It accurately measures if a sample falls within a class's true high-density region during Local View scoring, significantly enhancing non-linear expressiveness.
>
> **3. Practical Efficiency and Hyperparameter Tuning**
> To address your deployment concerns without redundancy, please refer to our responses to Reviewer y6R5:
> * **Inference Latency:** (See Reviewer y6R5, Weakness 1). We have compressed inference overhead to a minimal level. The optimized HC-SOINN is now faster than the KAC (CVPR 2025).
> * **Hyperparameters & Usability:** (See Reviewer y6R5, Weakness 2). Our method is practically "tuning-free" as we used the exact same set of hyperparameters across all diverse benchmarks. The performance leap stems from inherent topological manifold modeling, not meticulous parameter tweaking.
>
> **4. Utilizing Full Graph Topology for Classification**
> Currently, our score computation (Eq. 7) only uses node distances. Edges primarily serve neighborhood co-evolution during training. Utilizing full graph topology (e.g., via GNNs or centrality algorithms) inevitably introduces significant computational and memory overhead. As you noted, CIL scenarios are highly sensitive to inference latency. However, leveraging full topology could theoretically further enhance robustness against local noise. Inspired by your feedback, we will discuss "topology-based inference confidence calibration" as a promising Future Work direction.
>
> **5. Integrating the FC Layer**
> Initially, we worried that severe catastrophic forgetting would cause the original parametric classifier (FC layer) to suffer from "task-recency bias," distorting topological classification.
> To verify your hypothesis and explore the fusion boundary, we added an ensemble experiment on DualPrompt + HC-SOINN with a dynamic score fusion mechanism: $\text{Result} = (1-w) \times \text{HC-SOINN} + w \times \text{FC}$.
> As shown in Table-3 (anonymous link), the results are intuitive:
> * **Moderate Fusion is Beneficial ($w=0.3$):** Assigning a small weight to the FC layer improves performance (e.g., Split CIFAR-100 $A_{Last}$ increased from 85.63% to 86.14%). This shows a calibrated FC layer retains complementary discriminative knowledge that synergizes well with our topological features.
> * **Over-reliance is Harmful ($w=0.5$):** Increasing the FC weight noticeably degrades overall accuracy (e.g., Split CUB-200 $A_{Last}$ dropped to 85.24%, lower than without fusion). This confirms our initial concern: excessive FC involvement reintroduces severe catastrophic forgetting and task bias, masking the topological architecture's anti-forgetting advantages.
>
> In conclusion, moderately introducing FC knowledge aids classification, but the weight should be kept low. Thank you for this insightful suggestion. We will add the fusion strategy and its weight sensitivity to the revision.
>
> Thank you again for your constructive feedback!

---

> > ### Author Rebuttal · Reviewer_Dy6q · 2026-04-03
> >
> > I thank the authors for the detailed explanation of my questions and concerns.
> >
> > In the original review, the main concerns are on the intuition behind each part of the proposed methods, and the practical efficiency. In the rebuttal, the authors addressed these two point by discussing the intuitive meaning of each parts of the methods. I suggest the authors to include these points in the revision to make the paper more clear.
> >
> > Overall, I think this paper proposed a new method for Class-Incremental Learning, which is well-motivated and seems to work well in practical experiments, thus I'm welling to increase my score to 5.

---

> > > ### Author Response · Authors · 2026-04-03
> > >
> > > Dear Reviewer Dy6q,
> > >
> > > We would like to express our sincere gratitude to reviewer Dy6q for acknowledging our work and providing insightful comments. We will incorporate the discussed clarifications into the final manuscript to improve clarity and completeness.
> > >
> > > Thanks again for the time and effort in reviewing our work.
> > >
> > > Authors

---

### Official Review · Reviewer_wzjD · 2026-03-10

**Soundness:** 3
**Presentation:** 2
**Significance:** 3
**Originality:** 3
**Overall Recommendation:** 4
**Confidence:** 4

**Summary:**

This manuscript studies classifier design for class-incremental learning under the observation that practical CIL often violates the idealized point-wise neural collapse assumption. This manuscript argues that class features form manifolds rather than single collapsed points, proposes HC-SOINN to model each class with multiple topology-aware sub-prototypes plus a global center, and further adds STAR to track non-linear drift of these topological nodes over tasks.

**Compliance With Llm Reviewing Policy:**

Affirmed.

**Final Justification:**

I recommend maintaining my score of 4. The rebuttal successfully resolved my concerns by clarifying the theoretical link to boundary stability, storage budget flexibility, and inference latency. However, I am not raising the score because these additions primarily fix presentation gaps and validate basic soundness, rather than elevating the paper's fundamental novelty.

**Key Questions For Authors:**

1) The current analysis seems to bound representation error under drift, but it is less clear how this directly supports improved classification or reduced forgetting.
2) Clarify how many representative old samples are stored, whether each topological node corresponds to one stored sample, and compare this budget fairly with prior CIL methods.
3) Discuss whether the method can be simplified or accelerated for deployment.

**Limitations:**

yes

**Strengths And Weaknesses:**

Strengths:
1) This manuscript focus on a meaningful problem: the mismatch between single-prototype classifiers and realistic class manifolds in CIL.
2) The method is conceptually clear and reasonably original in how it combines hierarchical clustering, SOINN-based topology refinement, dual-view inference, and residual alignment.
3) The empirical evaluation is fairly broad, with consistent gains across multiple baselines and datasets, and the ablations help clarify the contribution of the main components.

Weaknesses:
1) The theoretical analysis is supportive but not fully aligned with the main claim. It bounds representation error under drift assumptions, but does not directly establish improved classification or reduced forgetting.
2) The inference cost increases noticeably compared with simpler classifier designs, which may limit the practical applicability of the method in real-world deployment scenarios.
3) The paper does not clearly specify the exact storage budget for representative old samples used by the method. This information should be reported more explicitly, ideally in comparison with prior methods, to enable a fair assessment of memory efficiency.

---

> ### Author Rebuttal · Authors · 2026-03-30
>
> Dear Reviewer wzjD,
>
> **(Note: For your convenience, all figures and tables mentioned in this response are available at the anonymous link: [https://anonymous.4open.science/r/ICML2026-Rebuttal-D2D6/](https://anonymous.4open.science/r/ICML2026-Rebuttal-D2D6/).)**
>
> We sincerely thank you for your "Weak Accept" recommendation and high evaluation. We have conducted further analysis regarding your crucial questions. Our detailed responses are below:
>
> **Response to Q1: Aligning Theoretical Analysis with "Improved Classification/Reduced Forgetting" Claims**
>
> We appreciate your insights and will explicitly connect our representation error bound to improved classification and reduced forgetting. HC-SOINN's superior performance over the single-prototype NCM is driven by two distinct mechanisms. **Our theoretical proof (Proposition 1) is specifically designed to mathematically guarantee the second mechanism.**
>
> * **1. Improved Classification via Manifold Fitting (Intra-task):** Multi-prototype methods naturally capture complex data manifolds more accurately than single-point assumptions, directly improving accuracy. Empirically, upon learning new classes, HC-SOINN achieves higher initial accuracy than NCM under the same backbone, showing superior representational power before forgetting.
> * **2. Reduced Forgetting via Boundary Stability (Inter-task - The Core Goal of Proposition 1):** In CIL, catastrophic forgetting occurs because feature drift non-linearly distorts old-class decision boundaries. Proposition 1 is established to prove that HC-SOINN is mathematically more resistant to this forgetting than NCM. It proves HC-SOINN's drift error bound is strictly smaller. **The direct logical link is: a smaller deviation in representation space mathematically guarantees less non-linear destruction of decision boundaries.** While a single prototype's shift drastically skews an entire class's decision half-space, multiple topological nodes disperse this global distortion into localized fine-tunings. Thus, Proposition 1's bounded representation error theoretically guarantees old-class boundary stability (i.e., reduced forgetting). We will add this logical bridge in our revision.
>
> **Response to Q2: Explicit Storage Budget for Old Samples and Fair Comparison**
>
> Thank you for pointing this out. We explicitly clarify: Yes, in the STAR module, each topological node corresponds exactly to one stored real historical anchor sample.
>
> Regarding the exact storage budget and fair comparison, we provided detailed analyses in Appendix D (Table 5) and E (Table 6). Specifically:
>
> * **Exact Storage Budget (Ref. Appendix D):** We analyzed the topological node count for various methods combined with HC-SOINN. Since STAR anchors strictly map one-to-one with nodes, this table represents our exact storage budget. Data shows on CIFAR-100 and CUB-200, the average stored samples per class for all baselines is strictly below 20 (~17-19). Only on ImageNet-R, due to massive intra-class variance, do some methods (e.g., DualPrompt, CODA-Prompt) slightly exceed 20 (overall average 20.61).
> * **Fair Comparison with Traditional Methods (Ref. Appendix E):** To evaluate memory efficiency, Appendix E directly compares our budget with classic rehearsal-based methods. Results show our method achieves significant performance gains over strong baselines while maintaining an identical or lower memory budget (e.g., averaging 18.48 samples/class on CIFAR-100).
>
> Furthermore, STAR's storage requirement is highly flexible. To demonstrate this elasticity, we added new validation experiments in the anonymous link (see **Table-2, Fig 1.4, and Fig 2.4**). In deployment, the STAR buffer size can be dynamically adjusted by tuning the target cluster count $K_{init}$ or SOINN refinement iterations $T_{soinn}$. Stored sample counts show a smooth positive correlation with classification accuracy (evident in **Fig 1.4 and Fig 2.4**).
>
> To prove this flexibility in strictly controlling the memory budget, we conducted a supplementary test for CODAPrompt and DualPrompt which slightly exceeding 20 nodes on ImageNet-R (see **Table-2**). By adjusting $T_{soinn}$ from 1 to 2, we successfully reduced their average node count strictly below 20 (e.g., DualPrompt dropped from 21.38 to 19.47). Notably, this further storage compression caused almost no performance loss and even slightly improved some metrics. This demonstrates our model can flexibly restrict the storage budget within specific thresholds based on edge device constraints while maintaining great accuracy. We will highlight this elasticity in the main text.
>
> **Response to Q3: Inference Cost and Deployment Acceleration**
> We understand your concerns. To avoid redundancy regarding our deep inference latency optimization and engineering acceleration details, **please refer to our response to Reviewer y6R5 under "Response to Weakness 1", specifically, "1.1 Inference Latency Optimization".**
>
> Thank you again for your constructive feedback!

---

> > ### Author Rebuttal · Reviewer_wzjD · 2026-04-03
> >
> > I thank the authors for their rebuttal to address my concerns. I would like to keep my positive score. Good luck.

---

> > > ### Author Response · Authors · 2026-04-03
> > >
> > > Dear Reviewer wzjD,
> > >
> > > We sincerely thank you for the thoughtful follow-up and constructive suggestions. We are glad that our clarifications have addressed your concerns. Thank you again for your time and effort in reviewing our work.
> > >
> > > Authors

---

### Official Review · Reviewer_y6R5 · 2026-03-10

**Soundness:** 4
**Presentation:** 3
**Significance:** 2
**Originality:** 2
**Overall Recommendation:** 4
**Confidence:** 4

**Summary:**

This paper proposes a topology aware hierarchical classifier named HC SOINN and an alignment module named STAR for class incremental learning. It challenges the single prototype assumption of Neural Collapse by modeling complex class manifolds and adapting to non linear feature drift, showing consistent improvements across multiple baseline methods and datasets.

**Compliance With Llm Reviewing Policy:**

Affirmed.

**Key Questions For Authors:**

Please see cons.

**Limitations:**

yes

**Strengths And Weaknesses:**

Pros

1.The motivation of this paper is well-grounded. The authors identify the limitations of Neural Collapse under practical class-incremental learning settings, and further propose a topology-aware approach by analyzing the structure of feature manifolds.

2.The proposed method is validated across multiple datasets and multiple baseline methods, demonstrating its broad effectiveness and generalizability.

Cons

1.While the authors claim in Section 6.1 that the training overhead is negligible, Table 3 clearly shows that inference latency increases by approximately 47%. Furthermore, maintaining multiple sub-prototypes and storing anchor samples for drift alignment inevitably increases memory consumption, which may become a bottleneck on resource-constrained edge devices.

2.The proposed framework introduces several additional hyperparameters, such as the balance factor between global and local views, the target number of clusters, and the maximum edge age. This may reduce the practical usability of the method when applied to real-world dynamic data streams.

---

> ### Author Rebuttal · Authors · 2026-03-30
>
> **Dear Reviewer y6R5,**
>
> **(Note: For your convenience, all figures and tables mentioned in this response are available at the anonymous link: [https://anonymous.4open.science/r/ICML2026-Rebuttal-D2D6/](https://anonymous.4open.science/r/ICML2026-Rebuttal-D2D6/).)**
>
> We sincerely thank you for your constructive feedback, your "Weak Accept" recommendation, and your recognition of our motivation and generalizability. Below, we address your concerns in detail:
>
> **Response to Weakness 1: Inference Latency and Memory Consumption**
>
> **1.1 Inference Latency Optimization:**
> We appreciate your rigorous check on efficiency. Acknowledging the initial latency overhead, we redesigned our inference pipeline with a "Two-Stage Coarse-to-Fine Retrieval" mechanism, which significantly reduces computational costs:
> * Instead of globally computing distances to all $C \times K$ sub-prototypes, we first rank all categories using NCM (class mean) distances. This requires only a single, lightweight $\mathbb{R}^{N \times C}$ matrix multiplication.
> * Next, we retain only the top-$k_c$ candidate classes per sample and compute sub-cluster distances exclusively for them. When $k_c \ll C$, the dominant $\mathbb{R}^{N \times M}$ computation decreases proportionally by roughly $C/k_c$.
> * Additionally, we performed comprehensive code-level optimizations, removing redundant CPU-GPU memory transfers and duplicate query normalizations in our PyTorch implementation.
>
> As shown in **Table-1** (anonymous link, based on CIFAR100 with CODAPrompt), our optimized implementation drastically reduces inference latency across two hardware platforms. While slightly slower than the minimalist NCM baseline, it is noticeably faster than the non-linear classifier KAC (CVPR 2025) and delivers higher accuracy. We believe this moderate trade-off is highly acceptable given the substantial performance gains.
>
> **1.2 Clarification on Memory Consumption:**
> Regarding concerns about memory overhead on resource-constrained edge devices, to avoid redundancy, **please refer directly to our detailed response to Reviewer wzjD's Question 2 ("Explicit Storage Budget for Old Samples and Fair Comparison")**.
>
> **Response to Weakness 2: Hyperparameters and Practical Usability**
>
> We understand the concern that additional hyperparameters might complicate real-world deployment. However, we wish to clarify that HC-SOINN is practically "tuning-free" across different tasks and datasets.
>
> * **Consistent Cross-Scenario Configuration:** Across all three diverse benchmarks (covering coarse-grained, fine-grained, and severe domain shifts) and all baselines, we strictly used an identical set of fixed hyperparameters ($\alpha=0.5$, $K_{init}=60$, $age_{max}=20$, $T_{soinn}=1$). Additionally, the STAR module's momentum $\lambda$ was uniformly set to 0.99 in all formal experiments (we apologize for omitting this in the appendix and will update it). We did not perform time-consuming grid searches for new tasks.
> * **Highly Robust Parameter Space:** To intuitively demonstrate this robustness, please refer to the comprehensive sensitivity analysis graphs in the anonymous link. **Fig 1.1 - 1.5** (CODAPrompt on CIFAR100) and **Fig 2.1 - 2.5** (SimpleCIL/DualPrompt on ImageNet-R) clearly show that performance remains highly stable across exceptionally broad parameter ranges.
>
> * **Unified Parameters Ensure Practical Usability:** Fine-tuning parameters for specific datasets can indeed sometimes yield higher performance. For example, as seen in **Fig 2.1** from the anonymous link, setting $\alpha$ to 0.6 achieves an $A_{Avg}$ of 70.43%, slightly higher than the 69.97% reported in our paper. However, these additional performance gains are not only limited in magnitude but are also often accompanied by trade-offs. For instance, as shown in **Fig 1.2** and **Fig 2.2**, although increasing the target cluster count $K_{init}$ improves accuracy, it comes at the cost of storing more topological nodes, which not only degrades inference efficiency but also exacerbates device memory bottlenecks. When dealing with constantly changing dynamic data streams in the real world, re-tuning parameters for new tasks is time-consuming. We intentionally report the results using a unified set of parameters because this configuration has proven its excellent practical usability across extensive benchmarks. The reason our model can handle various scenarios with a fixed configuration is that the topological mechanism relies on data-driven, self-organized growth. This dynamic representation of complex manifolds naturally possesses strong environmental adaptability, thereby alleviating the burden of manually fine-tuning hyperparameters.
>
> We will include these comprehensive sensitivity analyses in the revised appendix and state this cross-dataset hyperparameter stability and deployment convenience in the main text. We hope these optimizations and clarifications address your concerns and demonstrate our framework's practical value.

---

> > ### Author Rebuttal · Reviewer_y6R5 · 2026-04-05
> >
> > Thanks for the rebuttal. I will maintain my rating.

---

> > > ### Author Response · Authors · 2026-04-05
> > >
> > > Dear Reviewer y6R5,
> > >
> > > We sincerely thank you for your constructive suggestions. We are glad that our clarifications have addressed your concerns. Thank you again for your time and effort in reviewing our work.
> > >
> > > Authors

---

### Decision · Program_Chairs · 2026-04-30

**Decision:**

Accept (regular)

**Comment:**

This submission presents a well-motivated and practically relevant contribution to class-incremental learning. It challenges the single-prototype assumption underlying NCM-style classifiers and replacing it with a topology-aware hierarchical classifier that models class manifolds through multiple adaptive sub-prototypes. Reviewers, in general, agree that the empirical evidence is strong in the sense that the method is evaluated across multiple baselines and datasets, it integrates cleanly into existing systems, and yields consistent gains. Primary concerns are related to the gap between theory and the paper’s strongest claims, increased inference/storage overhead, and the practical complexity introduced by several hyperparameters. The rebuttal addresses these issues somewhat convincingly by clarifying the intended theoretical scope, giving more explicit insight into different parts, and adding intuition + additional analyses for the main design choices. All three detailed reviewers who raised concrete concerns marked them as fully resolved and maintained or increased positive scores. From my perspective, this warrants an accept decision, yet the authors should be careful not to overstate the paper's theoretical implications and should make efficiency tradeoffs more explicit in the final version.